

# Chemical composition, source and formation mechanism of urban PM₂.₅ in Southwest China

Junke Zhang[1], Yunfei Su[1], Chunying Chen[1], Wenkai Guo[1], Miao Feng[2], Danlin Song[2], Tao Jiang[3], Qiang Chen[3], Yuan Li[3], Wei Li[2], Yizhi Wang[2], Qinwen Tan[2], Ruohan Wu[1], Ruiyan Pu[1], Minhui Lu[1], Xuhui Shen[1], Xiaojuan Huang[4]

[1]Faculty of Geosciences and Environmental Engineering, Southwest Jiaotong University, Chengdu, 611756, China

[2]Chengdu Academy of Environmental Sciences, Chengdu, 610072, China

[3]Sichuan Academy of Eco-Environmental Sciences, Chengdu, 610041, China

[4]Department of Environmental Science & Engineering, Shanghai Key Laboratory of Atmospheric Particle Pollution and Prevention (LAP3), Fudan University, Shanghai 200438, China

*Correspondence to*: Junke Zhang (zhangjunke@home.swjtu.edu.cn) and Danlin Song (sdl@airmonster.org)



**Abstract.** Despite significant improvements in air quality in recent years, the Sichuan Basin (SCB) is still facing frequent haze pollution in winter. In this study, the chemical components of $PM_{2.5}$ in a typical

pollution period at the beginning of 2023 in Chengdu, a megacity in the SCB, were characterized by bulk-chemical and single-particle analysis, and the $PM_{2.5}$ sources and formation mechanism of pollution were analyzed. The average mass concentration of $PM_{2.5}$ during the study period was $95.4 \pm 29.7$ µg m$^{-3}$. Organic matter (OM) was the most abundant component (35.3%), followed by nitrate (22.0%), sulfate (9.2%) and ammonium (7.8%). The aerosol particles were classified into five categories: mineral, OM,

S-rich, soot and fly ash/metal particles, and most of them were in the state of internal mixing. The entire observation period could be divided into two non-pollution periods (NP-1 and NP-2) and two haze periods (Haze-1 and Haze-2). With the evolution of pollution, the bulk-chemical and single-particle analysis exhibited similar characteristics, suggesting that Haze-1 was mainly caused by pollutants related to fossil fuel combustion, especially mobile sources, while Haze-2 was triggered by the rapidly increasing

secondary pollutants. The $PM_{2.5}$ sources included dust (8.5%), biomass burning (3.5%), coal combustion (15.5%), industrial processes (6.5%), vehicular emissions (25.6%) and secondary sources (40.5%). Analysis of WRF-Chem model results showed that the average contributions of local sources and regional transmission to pollution in Chengdu were the same (50% vs 50%). In addition, the source composition and WRF-Chem simulation results in different periods confirmed our analysis of the

formation mechanisms of the two haze events.

This study confirms that, despite the significant reduction in pollution experienced by Chengdu in the past decade, further significant reductions in $PM_{2.5}$ are still needed, with particular emphasis on vehicular emissions and secondary sources. High intensity local emissions or large amounts of regional transmission may cause serious haze events, and more effective policies for local emissions reduction or

joint prevention and control of regional air pollution are necessary in the future.

## 1 Introduction

In the past few decades, large quantities of pollutants have been emitted during the rapid urbanization, socioeconomic development and associated increase in motorized vehicles in China, which have caused frequent haze events, particularly for some of the highly industrialized and densely populated regions,

such as North China, Pearl River Delta, Yangtze River Delta, Fenwei Plain and Sichuan Basin (SCB) (Zhang et al., 2017b; Ji et al., 2014; Liu et al., 2016; Zhu et al., 2016; An et al., 2019; Huang et al., 2014). $PM_{2.5}$ (i.e., particulate matter with an aerodynamic diameter less than 2.5 µm) is the most important species that causes haze, and comprises a complex mixture of species either emitted directly into the atmosphere (primary) or produced in the air via gas-to-particle conversion (secondary). Many studies

have pointed out that elevated levels of $PM_{2.5}$ can reduce visibility, adversely affect human health and ecosystems, and influence climate change directly by absorbing and reflecting solar radiation and indirectly by modifying cloud formation and properties (Pöschl, 2005; Seinfeld and Pandis, 2006; Group, 2016).

Accurate analysis of $PM_{2.5}$ sources and clarification of pollution formation mechanisms are considered

important prerequisites for formulating effective science-based policies (Huang et al., 2017; Wang et al., 2014; Zhang et al., 2020a; Wang et al., 2018; Zhang et al., 2020b). Therefore, much research on these scientific issues has been conducted, with extremely valuable results obtained. For example, it has been noted that stagnant meteorological conditions (e.g., a low wind speeds, high humidity and a shallow boundary layer), primary emissions (e.g., industry, households and vehicular exhaust emissions),



secondary formation (e.g., homogeneous or heterogeneous reactions) and regional air transport can initiate the rapid formation and persistent evolution of haze episodes in China (Wang et al., 2014; Huang et al., 2014; Song et al., 2022; Zhang et al., 2020b). Moreover, the two-way feedback between the accumulation of air pollutants and depression of the atmospheric boundary layer can also aggravate haze pollution (An et al., 2019; Quan et al., 2013; Zhong et al., 2019). The results of these studies have helped

guide the formulation of China's air quality improvement policies in recent years, such as the "Atmospheric Pollution Prevention and Control Action Plan" during 2013–2017 and the "Three-year Action Plan to Win the Blue Sky Defense War" during 2018–2020 (Zhou et al., 2022), and eventually led to a reduction in $PM_{2.5}$ mass concentrations in various regions across China. For example, the annual mean $PM_{2.5}$ concentration showed a sharp reductions from 89.5 µg m$^{-3}$ in 2013 to 38 µg m$^{-3}$ in 2020 in

Beijing, and the number of severely polluted days also decreased, by 83% from 58 to 10 days, during the same period (Zhou et al., 2022).

Because analyzing the sources and formation mechanisms of $PM_{2.5}$ pollution is intrinsically linked to its physicochemical characteristics, such as its chemical composition, particle size distribution and mixing state, a variety of methods have been used in this regard. Among these methods, bulk-chemical and

single-particle analysis are two important approaches used currently, which can characterize $PM_{2.5}$ from different perspectives. Bulk-chemical analysis approaches, such as filter sampling or using instruments like an aerosol mass spectrometer, aerosol chemical speciation monitor or monitor for aerosols and gases, can accurately determine the mass concentration of $PM_{2.5}$ and its chemical components (e.g., organic matter (OM), inorganic component and metals) (Sun et al., 2014; Zhang et al., 2014; Decarlo et al., 2006;

Wang et al., 2022a; Xu et al., 2018). Since the air quality standard is based on the mass concentration of pollutants (including $PM_{2.5}$), these quantitative measurements can be directly related to the evaluation of air quality and play a vital role in the air quality improvement process. However, bulk-chemical analysis approaches provide an overall analysis of the $PM_{2.5}$ samples collected and therefore miss some critical microscopic particle information such as their morphology and mixing state, which is important for

simulating and evaluating the impacts of $PM_{2.5}$ on climate and human health. Thinking specifically about the mixing state of $PM_{2.5}$, a change in it may cause a huge changes to its secondary effects (such as its optical properties, health effects, hygroscopicity or cloud condensation nuclei activity), even contrary to its initial state (Liu et al., 2021). For example, Zhang et al. (2008) found that coating with sulfuric acid and subsequent hygroscopic growth led to a tenfold increase in the light-scattering coefficient of black

carbon (BC) particles, and a near twofold increase in the light absorption coefficient at a relative humidity (RH) of 80%, as compared to uncoated BC particles. Fortunately, this information can be obtained by some single-particle analysis methods, such as the aerosol time-of-flight mass spectrometer, single particle aerosol mass spectrometer, transmission electron microscopy (TEM) or nanoscale secondary ion mass spectrometry (Zhang et al., 2020b; Gard et al., 1997; Li et al., 2011; Zhang et al., 2019). Due to the

complementarity of the bulk-chemical and single-particle analysis methods in the determination of the physicochemical characteristics of $PM_{2.5}$, an increasing number of studies have combined these two methods to study the haze processes. Doing so provides more detailed information on the evolution, formation mechanisms and sources of $PM_{2.5}$ pollution (Zhang et al., 2020b; Zhang et al., 2021; Dall'osto et al., 2009; Salcedo et al., 2010). For example, Zhang et al. (2020b) integrated filter sampling and TEM

methods to investigated the causes of two types of haze processes in northeast China and found that one type of haze process was mainly induced by the accumulation of primary OM and deteriorated by secondary aerosol formation, while the other type was caused by the long-range transport of agricultural biomass burning emissions.



Across the different regions or cities of China, haze formation mechanisms will differ because of the different characteristics of emissions and meteorological conditions (Wang et al., 2021). The SCB, located in southwest China, is not only one of China's most economically developed and industrialized regions, but is the area that most frequently experiences haze events. Chengdu is the largest city in the SCB, with a population of well over 20 million. Meanwhile, by the end of 2022, car ownership in Chengdu had exceeded 6 million, ranking second only to Beijing. Topographically, Chengdu is located

in the west part of the SCB, which is surrounded by the Qinghai–Tibet Plateau, Yunnan–Guizhou Plateau, Qinling–Daba Mountains and Wushan Mountains in the west, south, north and east, respectively (Fig. 1) (Peng et al., 2020). Air pollution is a serious issue for this city, presumably due to its large population, unfavorable atmospheric diffusion conditions and relatively high humidity. Although the concentration of $PM_{2.5}$ had reduced to 39.8 μg m$^{-3}$ in 2021, it still does not meet the first grade of the Chinese National

Ambient Air Quality Standard (NAAQS; annual average of 35 μg m$^{-3}$), and is eight times the World Health Organization (WHO) guideline value (annual average of 5 μg m$^{-3}$). At the same time, heavy haze pollution still occurs frequently in winter in Chengdu, which has a serious impact on the everyday lives of local residents. Despite numerous studies having used multiple methods to investigate the physical, chemical and seasonal characteristics of $PM_{2.5}$ during haze, our knowledge of its sources, evolutionary

processes and formation mechanisms is still incomplete. At the same time, recent research found that, with the reduction in pollution, atmospheric $PM_{2.5}$ in China has shown many new features, such as a higher nitrate contribution, enhanced atmospheric oxidizing capability and a stronger secondary source contribution (Feng et al., 2021; Geng et al., 2019; Huang et al., 2021; Song et al., 2022). This means that the formation mechanism of atmospheric haze pollution in Chengdu may also be different from previous

reports.

At the beginning of 2023, Chengdu experienced several severe haze events, during which the observed $PM_{2.5}$ mass concentration frequently exceeded the China's NAAQS. So far, the sources, evolutionary processes and formation mechanisms of these haze events remain unclear. Since the development of effective air pollution control policies rely on such knowledge, in-depth research on these heavy haze

events it is urgently needed. Accordingly, in this study, a continuous observation campaign at the beginning of 2023 was carried out in the field. The formation mechanisms of two haze events were analyzed based on bulk-chemical and single-particle approaches as well as air quality model simulations. Here, we report (1) the mass concentration of $PM_{2.5}$ and its chemical composition measured by the bulk-chemical analysis; (2) the number composition of particulate matter and its mixing state measured by the

single-particle method (TEM); and (3) the formation mechanisms of the two haze events by analyzing them via a combination of chemical component and source apportionment results along with model simulations.

## 2 Materials and methods

### 2.1 Observational site

The field campaign was performed at a Air Quality Super Observatory, which is located in Qingyang District in the center of Chengdu (30.65°N, 104.03°E). Sampling was conducted on the roof of a building (approximately 25 m above the ground) from 26 January to 7 February 2023. The site had no surrounding tall buildings within 200 m and is affected by multiple local emissions, including nearby restaurants, traffic and a variety of residential sources. Thus, the site is representative of a typical urban environment





in Chengdu.

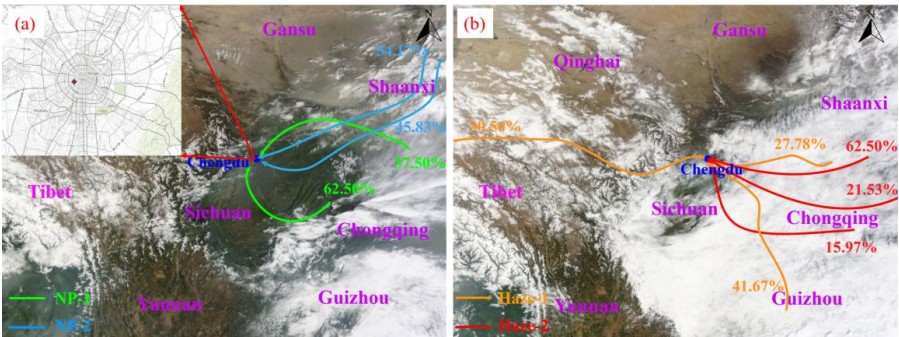

**Figure 1: Sampling site and the composition of air masses in different periods (the background maps of (a) and (b) are from https://worldview.earthdata.nasa.gov/, and the map of sampling site (the red dot) in the upper left corner of (a) is from https://map.baidu.com/).**

**2.2 Aerosol chemical components, air pollutants and meteorological parameters**

In this study, the mass concentrations of main chemical components of $PM_{2.5}$ (i.e., carbonaceous components, water-soluble inorganic ions and elements) were measured. The carbonaceous components, i.e., organic carbon (OC) and elemental carbon (EC), were measured with a semi-continuous OC/EC analyzer (Model 4, Sunset Laboratory, USA). The air was drawn through two quartz-fiber filters packed

together with a flow rate of 8 L min$^{-1}$, and $PM_{2.5}$ was collected onto a sampling spot of 1.31 cm$^2$. The collected sample was subsequently analyzed by the thermal–optical method. The cations ($Na^+$, $NH_4^+$, $K^+$, $Mg^{2+}$ and $Ca^{2+}$) and anions ($Cl^-$, $NO_2^-$, $NO_3^-$ and $SO_4^{2-}$) were measured with a In-situ Gas and Aerosol Composition Monitor (IGAC, Model S-611, Fortelice International Co., China), which is a semicontinuous monitor that separates gases and aerosols into liquid effluent for online chemical analysis

at an hourly temporal resolution. The IGAC system mainly consists of three components including a wet annular denuder, an aerosol collector and two ion chromatography systems. Air samples were pumped into the system at a flow rate of 16.7 L min$^{-1}$. Gaseous and aerosol samples were alternately collected, evacuated and analyzed with a temporal resolution of 30 min. The elements (K, Al, Si, Ca, Ti, V, Cr, Mn, Fe, Co, Ni, Cu, Zn, As, Cd, Ba and Pb) were monitored by an Xact 625 Ambient Metals Monitor (Cooper

Environmental Services LLC, USA). The ambient air was sampled on a Teflon filter tape through a $PM_{2.5}$ cyclone inlet at a flow rate of 16.7 L min$^{-1}$. Then the sample was automatically analyzed by nondestructive energy-dispersive X-ray fluorescence to determine the mass of metals. Detailed information about these instruments has been well documented elsewhere (Zheng et al., 2014; Huang et al., 2018; Young et al., 2016; Furger et al., 2017; Yu et al., 2018).

The hourly mass concentrations of several air pollutants ($PM_{2.5}$, $PM_{10}$, CO, $SO_2$, $O_3$ and CO) were downloaded from the official website of the China National Environmental Monitoring Centre (http://www.cnemc.cn/). Meteorological parameters, including air temperature (T), relative humidity (RH), wind speed (WS), wind direction (WD) and visibility, were obtained from the meteorological station (WS600-UMB, Lufft, Germany) of Sichuan Academy of Eco-Environmental Sciences,

approximately 3.5 km away from the air quality observation station.

In addition, in order to analyze the interannual evolution of pollution in Chengdu and its pollution level in China, we also compared the main meteorological parameters (T and RH, downloaded from



https://www.timeanddate.com/) and pollutants ($PM_{2.5}$, CO, $SO_2$, $O_3$ and CO, downloaded from
http://www.cnemc.cn/) observed in the same time period as this study (i.e., 26 January to 7 February) in
2015 and 2018 in Chengdu and in 2023 in some typical cities in various regions of China, including
Beijing (representing the North China Plain), Shanghai (representing the Yangtze River Delta),
Guangzhou (representing the Pearl River Delta) and Xi'an (representing the Fenwei Plain) (Table 1).
Unless otherwise specified, all online observation results in this study were presented at a 1-h resolution,
and expressed in Beijing standard time, which is 8 h ahead of coordinated universal time.

### 2.3 Single particle aerosol sample collection and analysis

Individual aerosol particles were collected onto copper TEM grids coated with carbon film (carbon type-
B, 300-mesh copper; Tianld Co., China) using a DKL-2 sampler (Genstar Electronic Technology Co.,
Ltd., China) with a single-stage cascade impactor equipped with a 0.5 mm diameter jet nozzle at a flow
of 1.0 L $min^{-1}$ in four periods (two non-pollution periods and two haze periods, see section 3.1.1).
Sampling times varied from 30 s to 3 min, depending on the particle loading as estimated from the
visibility. The collection efficiency of the impactor is 50% for particles with an aerodynamic diameter of
0.1 mm and a density of 2 g $cm^{-3}$. Individual particle samples were placed in a clean, airtight container
with controlled T (25 °C) and RH (20±3%) before being analyzed via TEM.
Individual particle samples were analyzed by a TEM at 200 kV accelerating voltage (JEM-2100, JEOL
Ltd., Japan) equipped with an energy-dispersive X-ray spectrometer (EDS, INCA X-MaxN 80T, Oxford
Instruments, United Kingdom). The morphology and mixing state of the aerosol particles were
determined by TEM, while EDS can detect elements with atomic weights corresponding to C and above.
Cu was not quantified because the Cu grids would have led to interferences. Detailed information on
individual particle analysis can be found in previous studies (Li et al., 2016; Deng et al., 2021). Ultimately,
a total of 1325, 1159, 995 and 1870 particles in four periods with a diameter < 2.5 μm were analyzed via
TEM-EDS.

### 2.4 Data analysis

#### 2.4.1 Chemical mass closure

According to previous studies (Huang et al., 2017; Zhan et al., 2023), the chemically reconstructed $PM_{2.5}$
mass ($PM_{chem}$) was calculated as comprised of eight categories, which can be expressed as follows:

$$[PM_{chem}]=[OM]+[EC]+[Cl^-]+[NO_3^-]+[SO_4^{2-}]+[NH_4^+]+[Mineral\ dust]+[Trace\ metals] \qquad (1)$$

In estimating OM, an OC to OM conversion factor of 1.6 was adopted for the aerosols at urban sites (Cao
et al., 2007; Turpin and Lim, 2001), i.e.,

$$[OM]=[OC]\times1.6 \qquad (2)$$

The calculation of mineral dust was performed on the basis of crustal element oxides. The Ti content was
very low (0.01 μg $m^{-3}$). Thus, eliminating the Ti content has an almost negligible influence on the
estimation of the mineral dust. Mineral dust was calculated as follows:

$$[Mineral\ dust]=2.14\times[Si]+1.89\times[Al]+1.40\times[Ca]+1.58\times[Mn]+1.43\times[Fe]+1.21\times[K]+1.67\times[Mg]$$
$$+1.35\times[Na] \qquad (3)$$

The trace metal reflects the sum of 12 different heavy metals and is expressed as:

$$[Trace\ metals]=V+Cr+Mn+Fe+Co+Ni+Cu+Zn+As+Cd+Ba+Pb \qquad (4)$$

Similar the results in Yangtze River Delta (Zhan et al., 2023) and Beijing-Tianjin-Hebei (Huang et al.,
2017) regions in China, we found that the $PM_{chem}$ concentration was lower than that of on-line $PM_{2.5}$



mass concentration ($PM_{on-line}$), the discrepancy between them can be considered as a chemical component that we did not measure and is defined as Unknown, i.e.,

$$Unknown = [PM_{on-line}] - [PM_{chem}] \tag{5}$$

### 2.4.2 WRF-Chem simulation

Air quality models are useful tools for explaining the formation and evolution of $PM_{2.5}$. The Weather Research and Forecasting model coupled with chemistry (WRF-Chem) belongs to a new generation of fully coupled regional meteorological air quality models using an "on-line" approach (Grell et al., 2005). In this study, we employed WRF-Chem to investigate the contributions of local sources and regional transmission during different periods in Chengdu. Specifically, we set up two simulation scenarios. The first was the baseline scenario, for which the model setup and validation of the scenario are detailed in the supporting information (S1 and S2); and the second was the external emissions scenario, in which we eliminated all anthropogenic emissions in the D03 region (Chengdu administrative area). The model settings for the two scenarios were consistent except for the emission differences. By subtracting the sensitivity scenario from the baseline scenario, the contributions of local sources and regional transmission to $PM_{2.5}$ could be obtained.

### 2.4.3 Backward trajectory analysis

The National Oceanic and Atmospheric Administration Hybrid Single Particle Lagrangian Integrated Trajectory (HYSPLIT) model has been widely used to simulate and analyze the movement, deposition and diffusion of airflow (Cohen et al., 2015; Draxler et al., 2009; Zhang et al., 2014; Luo et al., 2020). In this study, this model was used to simulate 48-h backward trajectories every hour for the four periods (see section 3.1.1) at the sampling site. The starting height for the back trajectories was 300 m. The reanalysis data with a spatial resolution of one degree and a temporal resolution of 1-h were obtained from the Global Data Assimilation System (GDAS) (https://rda.ucar.edu/datasets/, last access: 26 July 2023). To identify the pollutant characteristics in the different dominant transport patterns, cluster analysis was performed on the trajectories using HYSPLIT, and two to three clusters were identified according to the similarity in their spatial distributions (Fig. 1).

### 2.4.4 Sulfur oxidation rate and nitrogen oxidation rate

The concentrations of $SO_4^{2-}$ and $NO_3^-$ are related to the concentrations of their gaseous precursors (i.e., $SO_2$ and $NO_2$) and the conversion rate from the precursor to particulate pollutants. The sulfur oxidation rate (SOR) and nitrogen oxidation rate (NOR) are defined as (Ma et al., 2017)

$$SOR = n(SO_4^{2-})/[n(SO_4^{2-}) + n(SO_2)] \tag{6}$$

and

$$NOR = n(NO_3^-)/[n(NO_3^-) + n(NO_2)] \tag{7}$$

where $n$ is the molar concentration.

### 2.4.5 Positive Matrix Factorization (PMF)

The U.S. Environmental Protection Agency's PMF 5.0 software was used in this work to perform the $PM_{2.5}$ source apportionment. PMF is a multivariate factor analysis tool based on a weighted least-squares fit, where the weights are derived from the analytical uncertainty (Paatero and Tapper, 1994; Paatero and



Hopke, 2003). The best model solution is obtained by minimizing the residuals obtained between modelled and observed input species concentrations. The uncertainty was set to 5/6 the method detection limit (MDL) when the data were less than or equal to the provided MDL. When the concentration was

greater than the provided MDL, the calculation was defined as

$$\text{Uncertainty} = \sqrt{(\text{Error Fraction} \times \text{Concentration})^2 + (0.5\text{MDL})^2} \qquad (8)$$

To reduce the error, samples with missing values for individual species were excluded rather than replaced by the mean concentrations of the remaining observations. In this study, a total of 23 species were used in the model (OM, EC, $Na^+$, $Mg^{2+}$, $Ca^{2+}$, $Cl^-$, $NO_3^-$, $SO_4^{2-}$, $NH_4^+$, K, Al, Si, Mn, Fe, Co, Cu,

Zn, Cd, Ba, Pb, $NO_2$, $SO_2$ and CO). Detailed information on factor selection and determination can be found in our previous studies (Zhang et al., 2024; Huang et al., 2017; Huang et al., 2021).

**3 Results and discussion**

**3.1 Overall characteristics of the study period**

**3.1.1 Meteorological and pollution characteristics**

The T and RH are important factors that determine the formation and decomposition of the chemical composition of particulate matter (such as secondary inorganic and secondary organic components). Their ranges of variation across the whole study period were 5.0°C–17.3°C and 31%–84%, respectively, with corresponding average values of 11.0±3.0°C and 62.4±11.6%, which are significantly higher values than those observed in the same period in North China, where T is often below zero and RH is low, such

as the values of −0.4±5.1 °C and 31.6±15.6% in Beijing and 3.6±4.1 °C and 33.6±14.3% in Xi'an (Table 1). This obvious difference between the South and the North China inevitably leads to a the difference in the formation mechanism of air pollution in these two regions (Wang et al., 2021). As shown in Fig. S3, the diurnal variation of T and RH shows obvious inverse correlation, with their highest values appearing in the afternoon (16:00, 14.2°C) and morning (07:00, 71.3%), respectively. The wind

determines the horizontal removal of pollutants. As can be seen from Fig. 2, the WS in most of the study periods (97%) was lower than 1.5 m s$^{-1}$, and the average WS was only 0.5±0.4 m s$^{-1}$, which means that the meteorological conditions were stagnant highly unfavorable for the horizontal diffusion of pollutants.

**Table 1. Air quality and meteorological conditions of different cities in China in the same time period and the interannual changes in Chengdu.**

| | Beijing(2023) | Shanghai(2023) | Guangzhou(2023) | Xi'an(2023) | Chengdu(2015) | Chengdu(2018) | Chengdu(this study) |
|---|---|---|---|---|---|---|---|
| PM$_{2.5}$ (μg m$^{-3}$) | 32.6±29.2 | 30.0±14.4 | 31.2±9.5 | 103.7±80.8 | 79.4±43.3 | 71.7±20.9 | 95.4±29.7 |
| PM$_{2.5}$/PM$_{10}$ | 0.49 | 0.59 | 0.60 | 0.66 | 0.67 | 0.69 | 0.77 |
| NO$_2$ (μg m$^{-3}$) | 26.4±19.0 | 31.1±18.9 | 25.5±14.9 | 52.9±28.0 | 49.6±20.0 | 49.7±13.0 | 43.5±19.2 |
| SO$_2$ (μg m$^{-3}$) | 3.1±2.1 | 7.2±2.0 | 5.5±1.2 | 13.9±6.4 | 16.1±12.1 | 8.5±3.3 | 3.6±1.4 |
| O$_3$ (μg m$^{-3}$) | 41.5±21.0 | 59.3±24.5 | 53.2±37.1 | 37.9±28.9 | 16.6±18.8 | 38.5±19.9 | 44.1±34.5 |
| CO (mg m$^{-3}$) | 0.4±0.3 | 0.7±0.2 | 0.7±0.2 | 1.1±0.5 | 1.4±0.5 | 1.0±0.2 | 0.9±0.2 |
| T (°C) | −0.4±5.1 | 5.0±4.6 | 15.1±5.4 | 3.6±4.1 | 6.0±2.9 | 3.2±2.6 | 11.0±3.0 |
| RH (%) | 31.6±15.6 | 64.5±21.3 | 70.4±24.0 | 33.6±14.3 | 79.7±16.0 | 75.6±20.6 | 62.4±11.6 |
| WS (m s$^{-1}$) | 1.9±1.4 | 2.2±1.4 | 2.1±1.5 | 1.7±1.1 | 1.7±1.1 | 1.5±1.0 | 0.5±0.4 |

The average mass concentrations of gaseous pollutants, i.e. $NO_2$, $SO_2$, CO and $O_3$, were 43.5±19.2 μg



$m^{-3}$, 3.6±1.4 µg $m^{-3}$, 0.9±0.2 mg $m^{-3}$ and 44.1±34.5 µg $m^{-3}$, respectively (Table 1). Although most of the study period (61.5%) could be classified as pollution days (daily average of $PM_{2.5}$ higher than 75 µg $m^{-3}$), the concentration of $SO_2$ was far lower than the NAAQS (60 µg $m^{-3}$), as well as previous winters' concentrations in the same period in Chengdu, such as 16.1±12.1 µg $m^{-3}$ in 2015 and 8.5±3.3 µg $m^{-3}$ in

2018. This could be due to the energy policies introduced in China to mitigated $SO_2$ emissions, including the desulfurization of coal-fired power plant plumes, the decommissioning of coal-fired boilers in manufacturing facilities and small power plants and the conversion of domestic coal use to cleaner fuels (Fang et al., 2009). At the same time, the $SO_2$ concentration in this study was lower than observations in all other considered cities except Beijing in the same period (Table 1). Compared with the value of

49.6±20.0 µg $m^{-3}$ in 2015, the $NO_2$ concentration in this study was only 12.3% lower, which is much less than the reduction for $SO_2$ (77.6%). At the same time, the $NO_2$ mass concentration in this study was higher than in all other considered cities except Xi'an. From 2015 to 2023, the mass concentration of CO decreased by 35.7%, and the comparison results between cities are similar to $NO_2$—that is, its mass concentration found in this study was only lower than that in Xi'an. The $O_3$ mass concentration,

meanwhile, was lower than that of Guangzhou and Shanghai, but higher than that of the two northern cities of Beijing and Xi'an. At the same time, it shows a trend of increasing year on year, with an increase of 165.7% from 2015 to 2023. This significantly enhanced $O_3$ mass concentration is consistent with the increasing atmospheric oxidizing capacity reported in many regions of China in recent years, which has been pointed out to change the formation mechanism of haze in China during that period (Feng et al.,

2021; Wang et al., 2023). In addition, significant correlation between CO and $NO_2$ was observed, with a correlation coefficient ($r$) of 0.81, indicating they have common sources. Due to increases in the vehicle population and mileages covered by vehicles in China, the emissions of $NO_2$ and CO apportioned to vehicular emissions have increased. In particular, by the end of 2021, the number of motor vehicles in Chengdu had exceeded 6 million. This rapid increase in vehicle ownership is also an important reason

why the rate of reduction in $NO_2$ is lower than that of $SO_2$. Therefore, motor vehicles and the increasing atmospheric oxidizing capacity in Chengdu may have an important impact on local air pollution.

During the whole study period, the average concentration of $PM_{2.5}$ was 95.4±29.7 µg $m^{-3}$, which was 2.7 times the NAAQS (35 µg $m^{-3}$) and 19.1 times the WHO guideline value (5 µg $m^{-3}$), thus indicating how significant the haze was during the measurement period. Although the mass concentration of $PM_{2.5}$

decreased by 9.7% from 2015 to 2018, it had rebounded sharply by 2023, with an increase of 33.1%. At the same time, the result in this study is only lower than that of Xi'an in the same period, but 2.9, 3.2 and 3.1 times that of Beijing, Shanghai and Guangzhou, respectively. This directly reflects that Chengdu still faces huge challenges in air pollution control. The $PM_{2.5}/PM_{10}$ mass ratio in this study is the highest among all the cities listed in Table 1, and increased from 2015 to 2023, thereby highlighting the more

important contribution of fine particles in the basin environment to the overall concentration of atmospheric particles.



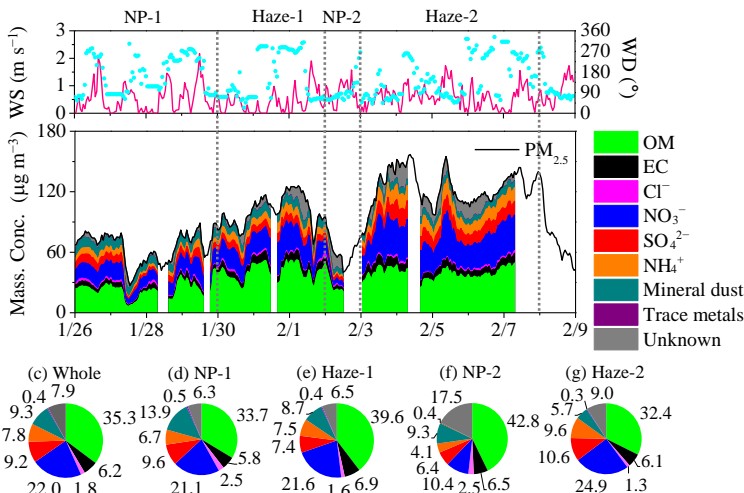

**Figure 2: Time series of (a) wind speed (WS) and wind direction (WD), (b) the mass concentration of PM$_{2.5}$ and its chemical components and (c–g) PM$_{2.5}$ chemical compositions in different periods (%).**

During the observation period of this study, Chengdu experienced multiple alternations between non-pollution and haze days. As shown in Fig. 2, there were two obvious haze events that occurred during the observation period. The first of these started in the early afternoon (13:00) on January 27. PM$_{2.5}$ rose slowly in a fluctuating manner with obvious accumulation characteristics. After 106 h, the PM$_{2.5}$ mass concentration reached its highest value of the first haze event, which was 125 μg m$^{-3}$. The PM$_{2.5}$ growth rate was 0.92 μg m$^{-3}$ h$^{-1}$. Subsequently, in the afternoon of February 1, short-term weak precipitation occurred. Meanwhile, the wind transformed into a stable easterly wind, and the WS was significantly higher than that in the period of high PM$_{2.5}$. Subsequently, the PM$_{2.5}$ mass concentration decreased rapidly. Therefore, this rapid reduction in PM$_{2.5}$ mass concentration was mainly caused by the precipitation and wind. It only took 31 h to reduce the maximum PM$_{2.5}$ concentration (125 μg m$^{-3}$) to the minimum (40 μg m$^{-3}$), and the rate of PM$_{2.5}$ reduction reached 2.74 μg m$^{-3}$ h$^{-1}$. However, at noon on February 2, with the decrease in WS and the disordered WD, PM$_{2.5}$ accumulated rapidly. In a short period of 37 h, the PM$_{2.5}$ mass concentration increased from 40 μg m$^{-3}$ to more than 150 μg m$^{-3}$. The growth rate reached 2.97 μg m$^{-3}$ h$^{-1}$. From February 4 to February 8, the mass concentration of PM$_{2.5}$ fluctuated but remained at a high level, with a daily average of more than 115 μg m$^{-3}$. Finally, the pollution process ended with the appearance of an easterly wind in the early morning of February 8. From the above description and the daily average value of PM$_{2.5}$ (NAAQS, 75 μg m$^{-3}$ for the 24 h average), the whole observation period can be divided into four stages: (1) non-pollution period 1 (NP-1), (2) Haze-1, (3) non-pollution period 2 (NP-2) and (4) Haze-2. The evolutionary characteristics and pollution formation mechanisms will be analyzed and discussed later.

In addition, it is worth noting that on 4 February, the daily and highest hourly PM$_{2.5}$ concentration reached 116 and 149 μg m$^{-3}$, respectively, leading to the "orange" haze alarm issued by the Government of Chengdu and the subsequent strict emission controls implemented during this period (http://sthj.chengdu.gov.cn/); for example, the time limit for motor vehicles was tightened, all open-air operations and all kinds of construction were prohibited and industrial enterprises were instructed to



cease production or had limitations imposed on their production depending on the type of goods they were producing. We then found that, in the next few days, although there was a significant increase in pollution at night, the pollution level in the daytime was significantly lower than that on February 3. This indicates that the various policies introduced to reduce pollutant emissions during the "orange" haze alarm period had a positive effect, which can be further demonstrated by the significantly reduced contributions of relevant pollution sources, such as industrial processes, vehicle emissions and dust (see section 3.5). However, the $PM_{2.5}$ mass concentration remained at a high level during this period, which means that there were other pollution sources or formation mechanisms strongly affecting the formation of haze, and so the coverage of the emission reduction policies during the alarm period was not sufficiently wide or stringent. Therefore, in order to further reduce similar levels of heavy pollution in future events, more comprehensive and powerful pollution reduction measures would be needed.

### 3.1.2 $PM_{2.5}$ chemical composition

As shown in Fig. 2c, OM constituted the largest component of $PM_{2.5}$, accounting for 35.3% of the total mass, and the contributions of three secondary inorganic species, i.e., $SO_4^{2-}$, $NO_3^-$ and $NH_4^+$ (SNA), were 9.2%, 22.0 % and 7.8%, respectively. Compared with previous winters' observation results in Chengdu, such as the order of contribution being OM (24.3%) > $SO_4^{2-}$ (15.1%) > $NO_3^-$ (12.9%) > $NH_4^+$ (9.0%) in January 2015, the contributions of OM and $NO_3^-$ were 11.0% and 9.1% higher, respectively, whereas the contributions of $SO_4^{2-}$ and $NH_4^+$ were 5.9% and 1.2% lower. This indicates the growing significance of OM and nitrate, as well as the diminishing significance of sulfate, in winter in Chengdu. This is consistent with the trends of change in their precursors; that is, the concentration of $SO_2$ has decreased year on year and been at a very low level. Although $NO_2$ has decreased in recent years, it is at a high level and frequently exceeds the NAAQS. In fact, due to the greater reduction in emissions of $SO_2$ than $NO_2$ and the negligible change in $NH_3$ since the implementation of China's Clean Air Action in 2013, a considerable increase in the nitrate fractions in aerosols has been observed during pollution events in most regions in China, and this the transition from sulfate-dominated to nitrate-dominated haze will likely have a significant impact on the mechanism of pollution formation, which is also worthy of attention when developing future pollution control measures (Huang et al., 2021; Geng et al., 2019; Wang et al., 2022a; Zhang et al., 2020a). For example, Xie et al. (2020) found that nitrate-rich particles can absorb more water than particles with higher sulfate fractions under moderately humid conditions (RH < 60 %), and that the particle pH increases rapidly owing to the combined effect of ammonia and nitrate, which will very likely occur in China in the coming years because both of these pollutants are not yet well controlled. Meanwhile, the changes in particle pH and hygroscopicity will further enhance the uptake of gaseous compounds and promote chemical reactions that favor lower acidity, as well as also affect the optical properties of airborne particles in China.

In addition, it is worth noting that the contribution of OM was significantly higher than observations in previous winters in Chengdu, such as in 2014 (27.8%), as well as in other cities in China, such as 14.5% in Beijing (Lv et al., 2022), 26.0% in Shanghai (Zhang et al., 2020a) and 23.6% in Xi'an (Wang et al., 2022b). In fact, our recent research on the interannual evolution of $PM_{2.5}$ in winter in Chengdu also found that the proportion of OM in $PM_{2.5}$ increased year on year, and its contribution to heavy pollution was also increasing (Zhang et al., 2024). Therefore, compared with other cities, the measures introduced to reduce emissions leading to air pollution in Chengdu needs to pay attention not only to the nitrate (similar to in other cities), but also to the OM with high pollution levels.

The mass ratio of $NO_3^-/SO_4^{2-}$ is used to indicate the relative importance of mobile and stationary sources



in the atmosphere. In this study, the $NO_3^-/SO_4^{2-}$ mass ratio during the whole study period was 2.5, which was significantly than that in winter 2011 (0.5) (Tao et al., 2014) and winter 2014 (1.1) (Wang et al., 2018) in Chengdu. Such an increasing $NO_3^-/SO_4^{2-}$ mass ratio is also apparent in other regions of China, especially in cities. For example, the $NO_3^-/SO_4^{2-}$ mass ratio in Beijing was 0.9 in 2011, which increased to 1.7 in 2020 (Wang et al., 2022a). This indicates that the role of mobile emissions has become increasingly significant owing to the rapid expansion of transportation.

Previous studies have suggested that when SOR and NOR are greater than 0.2 and 0.1, respectively, intense conversion and formation of secondary inorganic aerosols takes place (Yang et al., 2015). In this study, the average values of SOR and NOR during the whole observation period were 0.64 and 0.28, respectively, which means that a strong secondary generation process had occurred, and this value was higher than that in winter 2018 in Chengdu (SOR: 0.39, NOR: 0.13) (Song et al., 2022). This enhanced secondary generation process explains why the concentrations of the two species ($SO_4^{2-}$ and $NO_3^-$) were still at a high level in spite of the reduced concentrations of their precursors.

Overall, although the air quality in Chengdu has improved significantly in recent years, it still faces serious air pollution in winter, especially in terms of $PM_{2.5}$ and gaseous pollutants related to mobile sources (such as $NO_2$). Similar to in other cities, the increasing contribution of nitrate is particularly noteworthy. Moreover, the OM, is significantly higher than in other cities, which must not be ignored.

**3.2 Classification and mixing state of individual particles**

Based on the morphology and elemental composition of individual particles, the particulate matter in winter in Chengdu can be classified five major aerosol components: mineral, OM, S-rich, soot and fly ash/metal particles. Figure 3 shows the TEM images and EDS spectra of various particle types.



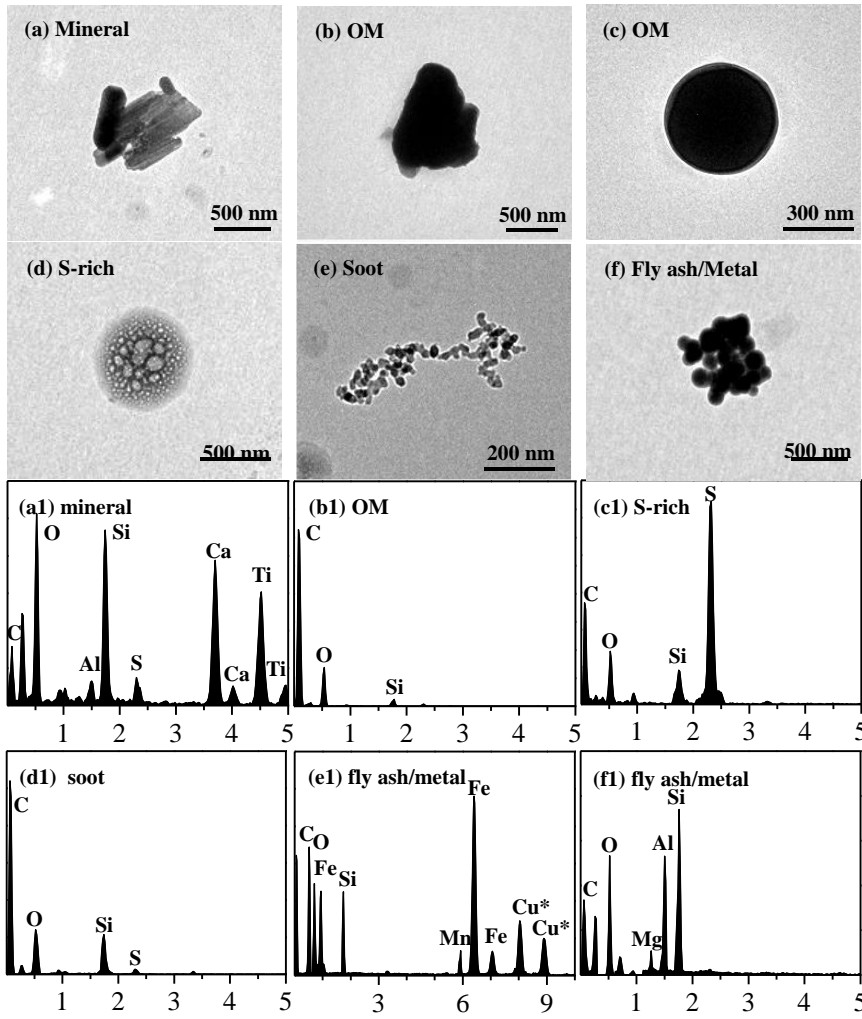

**Figure 3: (a–f) TEM images and (a1–f1) EDS spectra of individual particles.**

Mineral particles present an irregular shape (Fig. 3a and 4c), mainly contain the elements C, O, Al, Si, Ca and Ti (Fig. 3a1), and largely occurred in the coarse size range (> 1 μm). OM particles exist in various forms and can be further classified into irregular OM, spherical OM and OM coating, based on their

morphology (Fig. 3b, 3c and 4), and are mainly composed of C and O (Fig. 3b1). S-rich particles are mainly composed of C, O, Si and S (Fig. 3c1) and formed from the oxidation of $SO_2$, NOx and $NH_3$. S-rich particles normally represent the mixture of $(NH_4)_2SO_4$ and $NH_4NO_3$ (Li et al., 2016). Soot particles (i.e., BC or EC) present a chain-like morphology consisting of an aggregate of carbonaceous spheres with diameters from 10 to 150 nm (Fig. 3e, 4b and 4f). Soot particles mainly contain C and minor amounts

of O and Si (Fig. 3d1). Fly ash/metal particles, with spherical morphology (Fig. 3f, 4d and 4e), are mainly composed of C, O, Si and metallic elements (e.g., Al, Fe and Mn) (Fig. 3f1). Moreover, these particles mainly fall within the ultrafine size range (< 100 nm), and are considered tracers of coal combustion taking place as part of industrial activity or at power plants. For a detailed introduction to the various





types of particles, readers are referred to previous studies (Li and Shao, 2009; Li et al., 2016; Deng et al., 2021).

Numerous studies have shown that aerosol particles from different sources tend to internally mix with each other in the atmosphere as they age (Li and Shao, 2009; Li et al., 2014; Xu et al., 2020; Yuan et al., 2019). Accordingly, we found many different kinds of internally mixed particles; for example, OM or S-rich particles were mixed with almost all other types of particles, as the core or coating of particles. Meanwhile, mineral, fly ash/metal and soot particles were usually coated by OM or S-rich particles as the core of particles (Fig. 4). Therefore, based on particle types and mixing states, aerosol particles were further classified into nine groups: OM, soot, mineral, fly ash/metal, OM-S, OM-soot, OM-mineral, OM-fly ash/metal and OM-S-soot particles (Fig. 4).

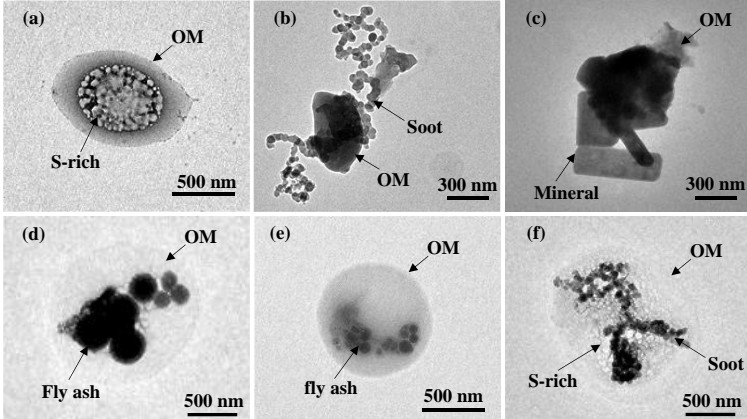

**Figure 4: TEM images of different types of internally mixed particles: (a) S-rich coated by OM; (b) mixture of OM and soot; (c) mixture of OM and mineral; (d, e) fly ash/metal coated by OM; and (f) mixture of OM, S-rich and soot particles.**

### 3.3 Exploring the formation mechanism of haze events

According to section 3.1, the study period was divided into four sub-periods: NP-1, Haze-1, NP-2 and Haze-2 (for ease, referred to simply as periods rather than sub-periods hereafter). This section comprehensively analyzes the changes in pollution characteristics during these four periods, with a focus on analyzing the formation mechanisms of the haze events, in order to provide scientific data that can be used to help formulate measures to reduce heavy pollution in winter in Chengdu.

#### 3.3.1 Meteorological conditions

Given the duration of NP-2 was only one day and the meteorological conditions on that day were characterized by rainfall and subsequent strong winds, this period was therefore an unconventional NP period, and its corresponding meteorological conditions were unsuitable for comparing with the results of the haze events to explore the formation mechanism of haze. Therefore, the comparison of meteorological conditions in this section mainly focuses on the two haze events and the more conventional NP period, i.e., NP-1. As shown in Fig. S4, the diurnal variations of T and RH in the three periods (NP-1, Haze-1 and Haze-1) were similar, i.e., with the highest and lowest T values appearing at 16:00 and 07:00–08:00, and the highest and lowest RH values appearing at 07:00–09:00 and 15:00–16:00,



respectively. However, the difference between the daily maximum and minimum values in each period was different, in which the T and RH in Haze-2 changed most smoothly, with the difference values being 3.9℃ and 12.2%, respectively, which was significantly lower than in the other periods—namely, 6.8℃ and 22.5% in NP-1, and 8.4℃ and 27.4% in Haze-1, respectively. In terms of their average values (Table 2), the T and RH of the haze events (11.5±2.9℃ and 61±9% in Haze-1, and 12.0±2.0℃ and 71±8% in Haze-2) were higher compared with those in NP-1 (8.7±2.7℃ and 57±12%), making it more conducive to the secondary generation of pollutants through photochemical and liquid-phase reactions. At the same time, the atmospheric pressure in the two haze periods (956 and 957 hPa in Haze-1 and Haze-2, respectively) was lower than in NP-1 (967 hPa), and the WS in the three periods were close and at a very low level (0.4–0.6 m s$^{-1}$). Overall, compared to NP-1, the meteorological conditions during the two haze events were more conducive to the accumulation and secondary generation of pollutants.

**Table 2. Meteorological conditions and pollutant indicators during different periods.**

|  | NP-1 | Haze-1 | NP-2 | Haze-2 |
|---|---|---|---|---|
| T (℃) | 8.7±2.7 | 11.5±2.9 | 14.1±1.9 | 12.0±2.0 |
| RH (%) | 57±12 | 61±9 | 50±6 | 71±8 |
| WS (m s$^{-1}$) | 0.5±0.5 | 0.4±0.4 | 0.7±0.4 | 0.6±0.4 |
| P (hPa) | 967±4 | 956±4 | 960±2 | 957±2 |
| CO (mg m$^{-3}$) | 0.8±0.2 | 1.1±0.2 | 0.7±0.1 | 1.0±0.2 |
| NO$_2$ (μg m$^{-3}$) | 32.4±15.4 | 61.5±18.2 | 31.3±14.2 | 42.8±14.5 |
| SO$_2$ (μg m$^{-3}$) | 3.0±1.3 | 4.0±1.6 | 3.0±0.4 | 3.6±1.5 |
| O$_3$ (μg m$^{-3}$) | 43.7±29.5 | 43.5±34.6 | 60.3±40.2 | 42.1±36.6 |
| NO$_3^-$/SO$_4^{2-}$ | 1.9±0.4 | 2.9±0.4 | 1.8±0.4 | 2.3±0.4 |
| NOR | 0.23±0.09 | 0.22±0.06 | 0.10±0.03 | 0.38±0.10 |
| SOR | 0.60±0.10 | 0.58±0.07 | 0.41±0.05 | 0.75±0.07 |
| PM$_{2.5}$ (μg m$^{-3}$) | 64.9±15.0 | 101.5±15.2 | 61.6±14.1 | 122.5±18.5 |
| PM$_{2.5}$/PM$_{10}$ | 0.67 | 0.77 | 0.70 | 0.83 |

It can be seen from Fig. 1 that the air masses during two NP periods originated from the east areas to Chengdu. The two clusters of NP-1 remained almost parallel before reaching Chengdu. All air masses originated from Hebei Province, then passed through Henan Province, Hubei Province and Shaanxi Province, and finally reached the sampling site from the east direction of Chengdu. The two clusters of NP-2 originated from Hubei Province and Chongqing, and finally reached the sampling site from the northeast and south directions of Chengdu, respectively. The three clusters of Haze-1 originated from the east, southeast and west areas to Chengdu, respectively, and there was no obvious turning point in the trajectory during transmission. All air masses during the Haze-2 period originated from the east areas to Chengdu and eventually reached the sampling site from the east and southeast directions of Chengdu. The difference in the direction of air masses during four peirods may imply that regional transmission has different impacts on pollution levels during different periods.

### 3.3.2 Gaseous pollutants

As shown in Table 2, compared with the NP periods, the main precursors, such as NO$_2$, SO$_2$ and CO, increased during the two haze events, which provided an important basis for the formation of PM$_{2.5}$ chemical components, such as nitrate, sulfate and OM. At the same time, O$_3$ concentrations during the two haze events were close to those of NP-1, and also showed obvious diurnal variations, with the highest values of Haze-1 and Haze-2 reaching 108.7 and 83.0 μg m$^{-3}$, respectively (Fig. S5). Thus,



photochemical reactions cannot be ignored in determining the mechanisms of formation of these heavy haze events. This is different to previous research findings for North China, where the concentration of $O_3$ during haze periods was found to decreased significantly and the contribution of photochemical reactions could be ignored (Lin et al., 2022; Yang et al., 2015).

### 3.3.3 PM$_{2.5}$ and individual particle composition

The average PM$_{2.5}$ mass concentration during the two NP periods was 64.9±15.0 μg m$^{-3}$ (NP-1) and 61.6±14.1 μg m$^{-3}$ (NP-2), which increased to 101.5±15.2 μg m$^{-3}$ (Haze-1) and 122.5±18.5 μg m$^{-3}$ (Haze-2) during the two haze events. Accordingly, the chemical composition of PM$_{2.5}$ also showed obvious differences with the evolution of pollution. As can be seen from the Fig. 2, from NP-1 to Haze-1, the contribution of the carbonaceous component showed the most significant increase, with OM and EC increasing by 5.9% and 1.1%, respectively. However, the mineral dust contribution showed the greatest decrease (by 5.2%), while the contributions of other components were similar during the two periods. The significant changes in chemical components suggest that the formation of this haze event was closely related to the combustion of fossil fuels. Meanwhile, compared to the NP-1, the significantly increased $NO_3^-/SO_4^{2-}$ mass ratio (from 1.9 to 2.9) and $NO_2$ concentration (61.5±18.2 μg m$^{-3}$, highest of the four periods) during the Haze-1 further suggests that the contribution of mobile sources—the most important fossil fuel combustion source in urban areas—may have dominated.

In the afternoon of February 1, short-term weak precipitation occurred. At the same time, the winds were generally easterly and the WS was higher than that in the period of high PM$_{2.5}$. Subsequently, the PM$_{2.5}$ mass concentration decreased rapidly, and correspondingly, the chemical composition of PM$_{2.5}$ changed significantly in NP-2. We found that, compared to Haze-1, the proportion of OM further increased by 3.2%, while the proportion of the three secondary inorganic components, i.e., SNA, decreased by 15.6%, with the largest decrease being for $NO_3^-$ at 11.2%. This is mainly because of the difference in hygroscopicity, i.e., the strongly hygroscopic SNA would have been more easily removed by precipitation, while precipitation would have been less efficient at removing the weakly hygroscopic, even hydrophobic carbonaceous components.

By the beginning of February 3, Chengdu had entered the Haze-2 period, featuring a higher PM$_{2.5}$ pollution level and longer duration. Compared with the NP-2 period, the proportion of OM decreased by 10.4%, while SNA increased by 24.2%, especially $NO_3^-$ (by 14.5%). This indicates that secondary generation was an important cause of Haze-2. Meanwhile, the $NO_3^-/SO_4^{2-}$ mass ratio and $NO_2$ also reached 2.3 and 42.8±14.5 μg m$^{-3}$, respectively, allowing us to infer that mobile sources and secondary generation jointly led to this more serious haze pollution event, with the latter possibly making the stronger contribution. The different contributions of secondary generation to the two haze events can be further confirmed by analyzing the SOR and NOR results, i.e., the two ratios in Haze-2 (0.75 and 0.38) were higher than those in the other periods, while their ratios in the Haze-1 period (0.58 and 0.22) were close to those in the NP-1 period (0.60 and 0.23).

The evolutionary characteristics of the chemical composition of individual particles as the pollution developed were similar to those revealed by the bulk-chemical analysis results. As shown in Fig. 5, from NP-1 to Haze-1, although the contribution of externally mixed OM particles decreased by 4.0%, the contribution of internally mixed OM particles, such as OM-S, increased by 5.8%. At the same time, the proportion of OM-soot, the internally mixed product of two types of carbonaceous particles, increased by 1.7%. Therefore, during this stage, not only did the proportion of OM-related particles increased, but so did the mixing of carbonaceous particles with secondary inorganic components. From Haze-1 to NP-





2, the occurrence of precipitation and sustained easterly winds reduced the total contribution of particles related to S-containing particles (including OM-S and OM-S-soot) from 46.8% to 19.7%. This would have been mainly due to the stronger hygroscopicity of S-containing particles, which determines their susceptibility to precipitation and strong winds. On the contrary, the contributions of weakly hygroscopic or hydrophobic carbonaceous particles, such as OM, soot and OM-soot particles, increased by 14.4%,

2.9% and 6.4%, respectively. When entering the Haze-2 period, the proportions of S-containing particles and internally mixed particles reached their highest levels among the four periods, with contributions of 62.1% and 70.0%, respectively. Therefore, in this period, particles experienced the strongest secondary aging and internal mixing. In addition, due to the limitations imposed by the pollution reduction measures during the "orange" haze alarm period on industrial sources, the total proportion of particulate matter

related to fly ash/metal (fly ash/metal and OM-fly ash/metal) decreased from 7.2% to 2.0%.

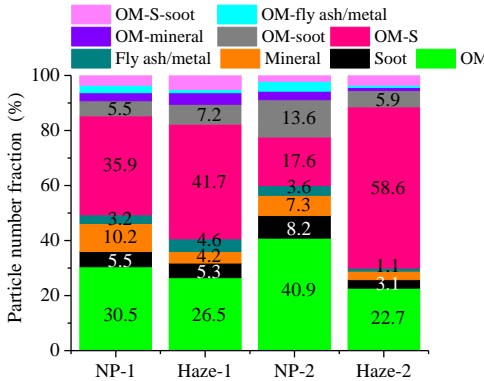

**Figure 5: Relative abundance of different particle types in different periods in Chengdu.**

## 3.4 Formation mechanism of secondary inorganic species

According to previous results (Huang et al., 2014; Huang et al., 2021; Zhang et al., 2014) as well as and

this study's results, we can state that secondary inorganic components are important chemical components that trigger haze events in Chengdu. Therefore, exploring their generation and mixing mechanisms is crucial to clarify the mechanisms of haze formation and pollution reduction. Previous studies have shown that photochemistry and liquid-phase reactions are the two important mechanisms for the generation of secondary inorganic components, and atmospheric oxidants (Ox=O$_3$+NO$_2$) and RH

are important indicators of these two processes, respectively (Sun et al., 2015; Wang et al., 2014). Therefore, in this section, we analyzed the relationship between these two factors and NOR and SOR, the results for which are shown in Fig. 6.



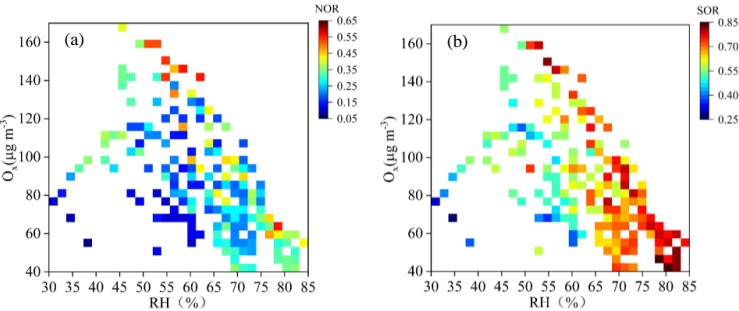

**Figure 6: The relationship between two secondary reaction indicators (Ox and RH) and (a) NOR and (b) SOR.**

As shown in Fig. 6, high NOR values were distributed in both the high Ox and high RH ranges, but their frequency and intensity were significantly stronger in the high Ox range than in the high RH range. Therefore, nitrate generation during the study period was mainly affected by photochemical generation. The high values of SOR were distributed in both the high Ox and high RH ranges, but their frequency and intensity were opposite to those of NOR, i.e., higher SOR values occurred more frequently in the high RH range. Therefore, the generation of sulfate during the observation period was mainly affected by liquid-phase generation.

From the morphologies of individual particles, it can be seen that secondary components mainly serve as the coating for various primary particles (Fig. 4). Many previous studies have demonstrated that these primary particles can provide surfaces for the heterogeneous reactions of $SO_2$, $NOx$ and volatile organic compounds (VOCs), which promotes the formation of secondary species in humid polluted air (Liu et al., 2021; Zhang et al., 2017a). For example, Liu et al. (2021) found that the presence of abundant burning-related POA particles could provide surfaces for heterogeneous reactions promoting high levels of production of secondary inorganic species particles, which would further elevated the pollution level.

It is worth noting that here that the secondary aging and mixing of particulate matter will play a positive feedback role in the aggravation of pollution (An et al., 2019). Ambient measurements in Beijing show that the secondary inorganic species fraction in $PM_{2.5}$ increases (from 24% to 55%) with increasing RH (from 15% to 83%), while the simultaneously elevated RH levels and secondary inorganic species mass concentrations result in an abundant aerosol liquid water content, which acts as an efficient medium for multiphase reactions and accelerates the formation of severe haze (Wu et al., 2018).

### 3.5 Source contributions

The source factors of $PM_{2.5}$ were apportioned by applying the PMF receptor model. The identification of the sources was based on certain chemical tracers that are generally presumed to be emitted by specific sources and are present in significant amounts in the collected samples. At the same time, it is necessary to consider the local pollution characteristics of Chengdu. Ultimately, six factors were identified in this study, and the source characteristics of all these factors are shown in Fig. 7. Because PMF has been widely used for $PM_{2.5}$ source analysis, the tracers for identifying the same source in previous studies are very similar (Huang et al., 2017; Huang et al., 2021; Srivastava et al., 2021). Therefore, the characteristics of each source are only discussed briefly here. Factor 1 was heavily weighted by $Na^+$, $Mg^{2+}$, $Ca^{2+}$, Al and Si, accounting for 41.1%, 67.2%, 50.9%, 65.8% and 49.8%, respectively, which was defined as the dust source. Factor 2 was mainly weighted by $K^+$ (51.0%) and $Na^+$ (26.3%), which were identified as



indicators of biomass burning. Factor 3 was identified by a high loading of $Cl^-$ (64.8%), Pb (46.5%) and $SO_2$ (42.9%), and a moderately loading of OM, EC, Cd, $NO_2$ and CO, which were regarded as signals of coal combustion. Factor 4 had a high abundance of Mn, Co, Zn and Pb, which are related to industrial

processes. Factor 5 can be considered as vehicular emissions, being mostly loaded with OM (32.9%), EC (42.9%), $NO_2$ (62.3%) and some metal elements released during the operation of motor vehicles, such as Mn, Fe, Cu and Zn. A high loading of $NO_3^-$ (51.9%), $SO_4^{2-}$ (59.6%) and $NH_4^+$ (57.6%), along with a moderate abundance of OM (30.6%), is apparent in Factor 6, which is typical of the secondary source profile.

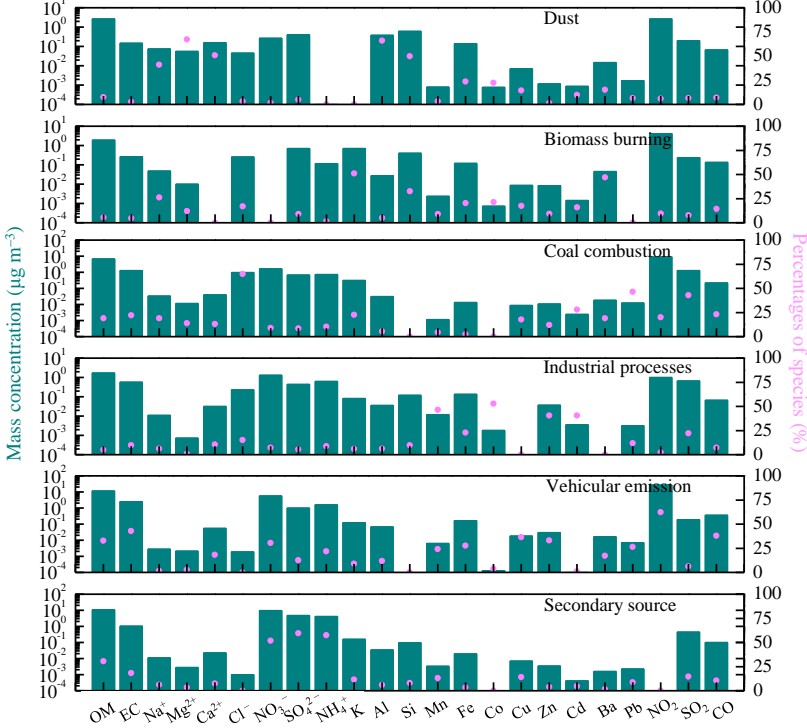

**Figure 7: PMF source profiles for PM$_{2.5}$ samples in terms of concentrations and percentages.**

As shown in Fig. 8, secondary sources were the highest contributor to PM$_{2.5}$ during the whole study period, accounting for 40.5%. According to this factor's source profile, it included secondary organic matter and secondary inorganic matter. Therefore, we can infer that secondary aerosols are produced by

625 secondary reactions of their precursors ($SO_2$, NOx, $NH_3$ and VOCs) in the atmosphere. Han and Zhang (2017) pointed out that secondary sources can contribute to regional transport in initial phases and heterogeneous reactions in later elevation phases. Vehicular emissions constituted the highest contributing factor among the primary sources, accounting for more than a quarter of the total PM$_{2.5}$ mass (25.6%). This high contribution was directly related to the level of vehicle ownership in Chengdu. In

addition, it is worth noting that non-exhaust vehicular emissions are an important source of OP$^v$ (the oxidative potential activity per volume of air of the aerosol component), an important indicator of the harm particulate matter causes to human health. Daellenbach et al. (2020) pointed out that, with the phasing out of older vehicles whose exhaust emissions are high in PM, non-exhaust emissions are



expected to constitute 80%–90% of the $PM_{10}$ derived directly from road transport after 2020. Accordingly, in the context of Chengdu's relatively high contribution of vehicular emissions currently, but with the relative contribution of exhaust emissions possibly decreasing in the future owing to the increasingly stringent vehicle exhaust emission standards and the rise in the proportion of new-energy (e.g., electric/hydrogen) vehicles, the contribution of non-exhaust emissions to vehicular emissions will become higher and pose a greater threat to human health. Meanwhile, the contribution of coal combustion (15.5%) was more than four times that of biomass burning (3.5%), which was mainly because the study period did not cover the traditional biomass burning season (late spring and early summer (Tao et al. (2013)); whereas, on the contrary, the pollutants emitted from coal combustion from certain industries (such as industrial furnaces and power plants) around Chengdu or other cities, may have had a serious impact on the air quality of Chengdu. In addition, the contribution of dust was 8.5%.

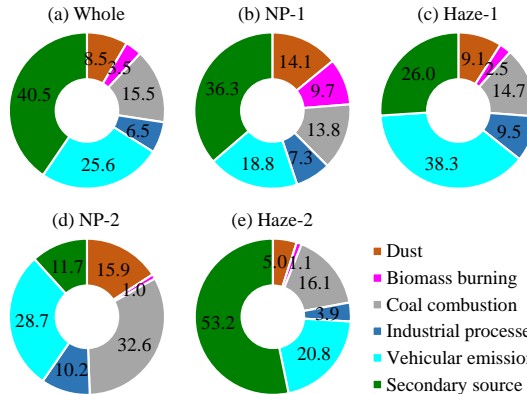

**Figure 8: Composition of PM$_{2.5}$ sources during different periods.**

Due to the impact of pollution reduction measures, urbanization and changes in pollutant formation mechanisms in recent years, the contributions of the various sources identified in this study were markedly different compared to previous winters. For example, the contributions of coal combustion, biomass burning and industrial processes in this study were lower than those reported in winter 2011 (18%, 16% and 18%, respectively) (Tao et al., 2014). This is mainly related to the various pollution reduction measures implemented in Chengdu and even the SCB in the past decade, such as the optimization of industrial infrastructure (e.g., factory renovations, relocations and closures) and the restructuring of energy (e.g., the replacement of bulk coal with clean energy), the improvements in industrial source pollutant emission standards (e.g., ultra-low emissions), the widespread use of various pollutant treatment technologies and the strict controls imposed on outdoor biomass burning in rural areas. The minimal reduction in coal combustion sources may be due to the rapid development of industry offsetting the effects of pollution reduction. At the same time, it is worth noting that although the contribution of coal combustion in this study is close to results reported in Beijing (13.3%) (Lv et al., 2022), compared to Beijing, there is no coal-fired heating in winter in Chengdu. Therefore, coal combustion in Chengdu still deserves attention. The contribution of vehicular emissions increased by 12.0% compared to 2018 (13.6%) (Song et al., 2022), and was significantly higher than that of other medium-sized cities in the SCB (8.6%-11.7%) (Zhang et al., 2023). This is mainly caused by the rapidly increasing number of motor vehicles. In addition, the contribution of secondary sources is close to the results in 2011 (44%) (Tao et al., 2014) and 2018 (43.4%) (Song et al., 2022) in Chengdu, and slightly





lower than that in winter in Beijing (47.7%) (Lv et al., 2022). Therefore, high levels of secondary sources are one of the common problems faced by many cities in China in reducing PM$_{2.5}$.

Figures. 8b–e show the PM$_{2.5}$ source compositions in the four periods. It can be seen that, from NP-1 to Haze-1, the contributions of secondary sources, biomass burning and dust decreased by 10.3%, 7.2% and 5.0%, respectively. According to previous studies, although local secondary generation or primary emissions contribute a certain amount to the particulate matter from these three sources, most of them mainly originate from regional transmission, especially biomass burning particles, which mostly originate from rural areas outside of Chengdu (Luo et al., 2020; Tao et al., 2013; Yang et al., 2012). This means that, compared to the NP1, the contribution from regional transmission weakened during Haze-1. On the contrary, the contribution of vehicular emissions increased by 19.5%; and meanwhile, the contribution from industrial processes—most of which derives from the suburbs of Chengdu—increased by 2.2%. During the NP-2 period, with the removal by the short-duration precipitation and strong easterly winds, the contributions of secondary sources and vehicular emissions decreased by 14.3% and 9.6%, respectively, while the contributions of coal combustion and dust sources increased by 17.9% and 6.8%, respectively. The continuous easterly winds carried pollutants related to coal combustion in eastern Chengdu and Chongqing, and greatly increased the contribution of the coal combustion factor. At the same time, this continuous easterly wind also drove the contribution of dust in NP-2 to reached its highest level across the four periods (15.9%). After entering the Haze-2 period, with the implementation of various emission reduction policies during the "orange" haze alarm period, compared with NP-2, the contributions of various primary sources decreased significantly. For example, the contributions of industrial processes, coal combustion and vehicular emissions decreased by 6.3%, 16.5% and 7.9%, respectively. With the decrease in wind power and vehicular emissions, the contribution of dust sources was also lowest among the four periods (5.0%). Contrary to the decrease in contributions from other sources, the secondary sources in the Haze-2 stage reached their highest value among the four periods (53.2%), which may be attributable to a large amount of secondary generation in Chengdu under the high humidity conditions and regional transmission (Lv et al., 2021). In fact, the source composition of PM$_{2.5}$ and its chemical components exhibited very similar characteristics during the evolution of the four periods, i.e., Haze-1 was mainly caused by the gradual accumulation of pollutant emissions from sources related to fossil fuel combustion, especially mobile sources, while, Haze-2 was mainly triggered by the rapid secondary generation. Even other sources also contributed to the formation of these two haze events to varying degrees.

### 3.6 Contribution of local sources versus regional transmission

Figure 9 shows the relative contributions of local sources and regional transmission in the WRF-Chem model results during the whole study period and different pollution period. It can be seen that, during the whole study period, the contributions of local sources and regional transmission were the same (50% vs. 50%). This means that pollution control in Chengdu still needs to involve strict joint prevention and control of regional air pollution on the basis of local pollutant emissions reduction. In terms of different periods, during the NP-1 period, the contribution of regional transmission reached its highest level among all the periods (56%). This means that, during conventional NP periods, compared to local sources, regional transmission has a more important impact on the pollution level of Chengdu. It is worth noting that, although the PM$_{2.5}$ level during the NP-1 period was lower than the Chinese NAAQS (24-h average of 75 μg m$^{-3}$), it still exceeded the WHO guideline value by more than four times (24-h average of 15 μg m$^{-3}$). This means that if Chengdu's air quality is to achieve further improvement after reaching the current





NAAQS, it may need more attention paid to the contribution of regional transmission. In terms of the two haze events in this study, the contribution of regional transmission during Haze-2 was higher than that during Haze-1, with contribution ratios of 52% and 40%, respectively. This further confirms our conclusion that local sources (such as mobile sources) were an important triggering factor for Haze-1, while regional transmission was key for the formation of Haze-2 (secondary pollutants). In addition, it is worth noting that during the Haze-2 period, Chengdu and surrounding areas adopted control measures during the "orange" haze alarm period. However, due to the lack of timely source information, we may have overestimated the contribution of local sources, and accordingly, the actual contribution of regional transmission during this period may have been higher. In addition, compared with simulation results during heavy haze periods in other cities in China, the contributions of regional transmission during Haze-2 are lower than those in Beijing (56%) (Li and Han, 2016) but higher than those in Shanghai (37%) and Suzhou (44%) (Li et al., 2015). This further highlights the differences in pollution formation mechanisms in different regions of China.

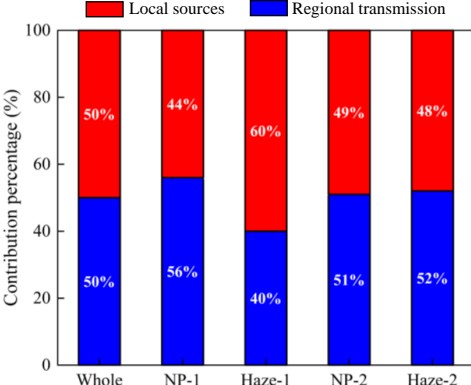

**Figure 9: Relative contributions of local sources and regional transmission to PM$_{2.5}$ during different periods.**

**4 Summary and conclusions**

The aerosol mass and chemical composition, sources and evolutionary processes at the beginning of 2023 in Chengdu were investigated with bulk-chemical and single-particle analysis along with numerical model simulations. Although the air quality in Chengdu has improved significantly in recent years, it still faces serious air pollution in winter, especially PM$_{2.5}$ (95.4±29.7 μg m$^{-3}$) and gaseous pollutants related to mobile sources (such as NO$_2$). Compared with other cities in China, PM$_{2.5}$ reduction in Chengdu not only needs to focus on the increasing contribution of nitrate, but also needs more attention paid to the high levels of OM. Based on the morphologies and elemental compositions of individual particles, these particles could be classified into mineral, OM, S-rich, soot and fly ash/metal particles, with most existing in internally mixed form in the atmosphere.

The whole study period included two non-pollution periods (NP-1 and NP-2) and two haze periods (Haze-1 and Haze-2). As pollution evolved, the chemical composition results obtained based on bulk-chemical and single-particle analysis exhibited strong consistency, i.e., from NP-1 to Haze-1, the mass contribution of OM and EC increased by 5.9% and 1.1%, respectively, and the number contribution of



OM-S and OM-soot particles increased by 5.8% and 1.7%. Meanwhile, the $NO_3^-/SO_4^{2-}$ mass ratio
reached its highest value across the four periods (2.9). The short-term precipitation and sustained easterly
winds led to a significant decrease in secondary inorganic species, i.e., SNA (by 15.6%) and S-containing
particles (by 27.1%). As pollution evolved from NP-2 to Haze-2, the contribution of SNA and S-
containing particles experienced explosive growth, with growth rates of 24.2% and 42.4%, respectively.
Meanwhile, the SOR and NOR both reached their highest values across the four periods (0.75 and 0.38,
respectively). It can therefore be inferred that the Haze-1 process mainly originated mainly from the
accumulation of fossil fuel combustion, especially mobile sources, whereas Haze-2 was triggered by the
large generation of secondary pollutants. The implementation of various emission reduction policies in
recent years has had a significant impact on $PM_{2.5}$ sources. Among them, the contributions of biomass
burning, coal combustion and industrial processes has decreased, while the contribution of vehicular
emissions has significantly increased. The WRF-Chem model analysis showed that local sources and
regional transmission make the same contribution. In addition, the differences in the sources and the
comparison between local sources and regional transmission across the four periods further validated the
different formation mechanisms of the haze events.

**Data availability.** The data used in this study are available from the corresponding author upon request
(Junke Zhang via zhangjunke@home.swjtu.edu.cn).

**Author contribution**: JZ and DS planned this campaign; JZ wrote the paper and led this research; YS,
CC and WG performed the data analysis and wrote the manuscript together with JZ; MF, TJ, QC, YL,
WL, YW and QT conducted experiments and instrument maintenance; RW, RP, ML, XS and XH helped
with the data analysis. All authors reviewed and edited the manuscript.

**Competing interests**: The authors declare that they have no conflict of interest.

**Disclaimer.** Publisher's note: Copernicus Publications remains neutral with regard to jurisdictional
claims in published maps and institutional affiliations.

**Acknowledgements.** We would like to thank Analysis and Testing Center of Southwest Jiaotong
University for the technical support in the single particle sample determination.

**Financial support.** This research has been supported by the National Natural Science Foundation of
China (grant no. 41805095 and 42205100) and the Basic Research Cultivation Support Plan of Southwest
Jiaotong University (2682023ZTPY016).

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
