# Peer review of "Chemical composition, sources and formation mechanism of urban PM2.5 in Southwest China: A case study at the beginning of 2023"

_EGUsphere, 2023_

## Referee Comment (RC1)

**Comments**

In the paper entitled "Chemical composition, source and formation mechanism of urban $PM_{2.5}$ in Southwest China", the authors used different techniques to investigate the properties of fine aerosols in Chengdu at the beginning of 2023. The investigated time period was divided into two pollution events and the reasons causing these pollution events were clarified. Although this paper showed us many pieces of information about the pollution, my feeling is that this paper is more like a report than a scientific paper. The methods used in this study are normal. The results are plenty but I did not find many new scientific findings from this paper. However, considering the hard work made in this study, I would like to give the authors a chance to revise their paper by highlighting the new findings in the revised manuscript. Thus, my suggestion is a major revision for this paper. The detailed comments are listed below.

1. Blank lines should be added between paragraphs.

2. Line 146, as the author mentioned, aerosols in China show new features recently. However, I did not find many new features presented in this paper. The one I found is that a less importance of sulfate and a stronger importance of nitrate in Chengdu. But what is the situation in other cities of China? I guess the situations are perhaps the same due to the policy released by the national government. Thus, I suggest the authors summarize the major new findings they found in this study and highlight them explicitly in the revised manuscript.

3. Line 231, please explain OM and EC here so that readers who are not familiar to these abbreviations can understand them.

4. Line 242, similar "to" the ….

5. Line 257, local sources include anthropogenic and natural emissions. The authors made a sensitivity test by switching off only anthropogenic emissions, so the results can only depict the influence of local anthropogenic sources

6. Lines 295-345, as the information given here is not very related with the topic of the paper, I suggest the authors largely shorten it.

7. Line 378, lower than that of Feb. 3 or Feb. 4? It was written that a haze alarm was released on

Feb. 4.

8. Lines 378-382, the reasons can be various such as the change of meteorological conditions, so please give your evidence here.

9. Table 2, is that better to show a figure instead of the table here as the numbers may not be that important.

10. Line 495, should we look at Fig. 1(a) or Fig. 1(b)?

11. Line 698, how did you calculate these contributions? Please give the definitions of the contributions and the methods obtaining them.

12. Line 698, again, the local sources here include only anthropogenic sources.

13. From Fig. S2 in the supplementary information, the model results of $PM_{2.5}$ and observations are not in a good consistency. Therefore, I doubt about the validity of the results shown in Section 3.6. In addition, my feeling is that the WRF-Chem simulation is not very associated with the topic of the paper, is that possible to remove this part?

14. Line 755, it is always better to share the data to the public instead of giving this sentence.

---

## Referee Comment (RC2)

In this study, the authors investigated the characteristics of aerosol chemical compositions by bulk-chemical and single-particle observations, analyzed the sources and formation mechanism of $PM_{2.5}$ pollution at the beginning of 2023 in Chengdu. The results of this observation experiment, including chemical components, meteorological conditions and source analysis, are comprehensive introduced in this paper. However, with the lack of the innovation in methods or ideas, this work did not provide significant new insights and scientific highlights. I think this paper is more a presentation of observational data rather than a scientific analysis as research article. In addition, the observation period is only about half a month, which does not represent the pollution level in southwest China as described in title. I also agree with previous reviewers of this article about the lack of analysis and discussion of new scientific findings. As a result, I would encourage the authors to carry out in-depth analysis and innovative results with major revision. But much further work is still required at this stage.

L171-174: The diagram in the upper left corner of Figure 1(a) needs to remove the information of road network.

L217: Please add references about density of 2 g cm$^{-3}$.

L372: Add the definition of "orange" haze alarm.

L495-505: This description seems superfluous and does not fit the main idea of the article.

L556-558: The particulate removal process is complex and there is no direct indication that this is influenced by the hygroscopicity here.

L605-620: Figure 7 and its associated descriptions are best moved to the supplement file.

L698-704: How were the contributions of local sources and regional transmission calculated?

L633: Please explain what is meant by "non-exhaust emissions".

The grammar of the essay needs a thorough examination.( for example, L125: "investigate" ; L257: "sensitivity"; L315: "mitigate"…)

---

## Author Comment (AC1)

Dear editor and reviewer,

Thank you very much for the comments and suggestions, which contribute to improve the quality of our paper. We have replied all comments and suggestions in our point-by-point response attached below. In order to highlight the changes what we have done, the color of the revised text will become blue.

**Response to Anonymous Referee #1**

**RC1:** In the paper entitled "Chemical composition, source and formation mechanism of urban PM2.5 in Southwest China", the authors used different techniques to investigate the properties of fine aerosols in Chengdu at the beginning of 2023. The investigated time period was divided into two pollution events and the reasons causing these pollution events were clarified. Although this paper showed us many pieces of information about the pollution, my feeling is that this paper is more like a report than a scientific paper. The methods used in this study are normal. The results are plenty but I did not find many new scientific findings from this paper. However, considering the hard work made in this study, I would like to give the authors a chance to revise their paper by highlighting the new findings in the revised manuscript. Thus, my suggestion is a major revision for this paper. The detailed comments are listed below.

Response. We are very grateful to the reviewer for giving us the chance to make revisions. Based on the reviewer's comments, we have made significant revisions to the structure and content of the manuscript. These revisions can be summarized as follows: (1) We have simplified some content that lacks innovation (section 3.1.1) or placed it in supplementary materials (Table 1 and the introduction to PMF factors) to ensure that the main text presents more important scientific information. (2) We conducted a more in-depth analysis of the obtained results: (a) when analyzing the regional transmission of pollutants, we not only analyzed the air mass transmission during different pollution periods, but also added the analysis results of concentration-weighted trajectory (Line 406-426), which can directly reflect the concentration contribution of regional transmission to the observation station; (b) we analyzed the sources and mixing structures of individual particles in the two haze periods, and

provided more scientific information and evidence for analyzing their formation mechanisms (section 3.3.4); (c) in order to provide readers with a clearer understanding of the evolution characteristics of pollution during the observation period, we have drawn a conceptual model of pollution evolution (Fig. 12), which displays the evolution characteristics of various meteorological and pollution indicators, particle mixing states and sources, and pollution formation mechanisms in the study period; (d) we discussed the significance of this study, particularly the importance of the TEM-EDS results in studying particle health and climate environmental effects (section 4.2). (3) We optimized the figures of the manuscript, such as presenting the content of Table 2 in the original manuscript in the form of figures (Figs. S5 and 7) and adding a conceptual model figure of pollution evolution (Fig. 12). These revisions ensures that the study results are more clearly expressed. We believe that through this major revision, the manuscript can present more and in-depth scientific information, allowing readers to better understand the sources and formation mechanisms of haze pollution in Chengdu. Thanks again for giving us the opportunity to make revisions.

1. Blank lines should be added between paragraphs.

Response. The formatting of the paragraph has been reset, i.e., blank a line before and after a paragraph.

2. Line 146, as the author mentioned, aerosols in China show new features recently. However, I did not find many new features presented in this paper. The one I found is that a less importance of sulfate and a stronger importance of nitrate in Chengdu. But what is the situation in other cities of China? I guess the situations are perhaps the same due to the policy released by the national government. Thus, I suggest the authors summarize the major new findings they found in this study and highlight them explicitly in the revised manuscript.

Response. Thanks for this important comment. We fully agree with the reviewer's comment that the original manuscript seriously lacks the innovative summary of this study, and the analysis of the data is not in-depth enough. Some of the results we reported, such as stronger nitrate contribution and weaker sulfate contribution, not only appeared in Chengdu but also in other cities in China. Therefore, during this revision process, we made significant revisions to the structure, content and presentation of the

manuscript. The revised manuscript provides a more in-depth analysis of the observed results, particularly by integrating the results of bulk-chemical and single-particle analysis, PMF results and WRF-Chem model to comprehensively analyze the formation mechanisms of the two pollution events. At the same time, we not only conducted a more in-depth analysis of the TEM-EDS results (section 3.3.4), emphasizing the importance of these results for the study of secondary effects of atmospheric particulate matter, but also pointed out that the integration of multiple methods is necessary for future air pollution research (section 4.2). To our knowledge, this study is the first to integrate these methods into the study of atmospheric pollution in Southwest China.

3. Line 231, please explain OM and EC here so that readers who are not familiar to these abbreviations can understand them.

Response. Thanks for this important comment. We have already explained OM and EC when they first appeared (Line 68 and Line 131).

4. Line 242, similar "to" the ….

Response. Corrected (Line 188).

5. Line 257, local sources include anthropogenic and natural emissions. The authors made a sensitivity test by switching off only anthropogenic emissions, so the results can only depict the influence of local anthropogenic sources

Response. Thanks for this valuable comment. We have added necessary explanations in the main text to ensure that readers can more accurately understand our research results (Line 205-206), i.e., "It is worth noting that this study mainly focuses on the contribution of anthropogenic sources, without considering the contribution of natural emissions, such as biogenic sources."

6. Lines 295-345, as the information given here is not very related with the topic of the paper, I suggest the authors largely shorten it.

Response. We fully agree with the reviewer's comment that this section is only a comparison of the

obtained results, reflecting very little scientific information and not very relevant to the research topic. Therefore, we have largely shortened this section (reduce from 971 to 391 words) and the Table 1 has been placed in the supplementary materials.

7. Line 378, lower than that of Feb. 3 or Feb. 4? It was written that a haze alarm was released on Feb. 4.

Response. We would like to explain the haze alarm policy in Chengdu, which means that based on the current level of pollutant emissions and meteorological conditions in the coming days, the environmental protection department can predict the air quality in the coming days. If it is predicted that severe pollution may occur in the next few days, the environmental protection department will release a haze alarm. At present, there are three levels of haze alarm in Chengdu:

**"Yellow" haze alarm**: It is predicted that the daily average of air quality index (AQI) > 200 (or $PM_{2.5}$ mass concentration >115 μg m$^{-3}$) will last for 2 days (48 hours) or more, and did not meet higher level warning conditions.

**"Orange" haze alarm**: It is predicted that the daily average of AQI > 200 for 3 days (72 hours) or more, or $PM_{2.5}$ concentration >115 μg m$^{-3}$ for 3 days (72 hours) or more, and $PM_{2.5}$ concentration >150 μg m$^{-3}$ for 1 day (24 hours) or more, and did not meet higher level warning conditions.

**"Red" haze alarm**: It is predicted that the daily average of AQI > 200 for 4 days (96 hours) or more, and that the daily average of AQI > 300 for 2 days (48 hours) or more; or it is predicted that the daily average of AQI will reach 500.

In order to reduce the harm of haze to the public, government departments will take corresponding pollution reduction measures after releasing a haze alarm. For example, in this study, after releasing the "orange" haze alarm, a large number of pollution reduction measures were taken in Chengdu and surrounding areas, such as the time limit for motor vehicles was tightened, all open-air operations and all kinds of construction were prohibited and industrial enterprises were instructed to cease production or had limitations imposed on their production depending on the type of goods they were producing. Under the influence of these emission reduction measures, the concentration level of pollutants during

the alarm period usually does not significantly increase, and may even decrease. Therefore, it is reasonable that the PM$_{2.5}$ concentration during the orange haze alarm was lower than that during the peak pollution period before its release.

We can simply understand that after releasing a haze alarm, if no pollution control measures are taken, there will definitely be more serious pollution in the next few days. The fact is, in order to prevent the occurrence of such heavy pollution, government departments will take a large number of pollution reduction measures after the haze alarm is released. Then, many key sources of air pollution (such as motor vehicles, industry, combustion processes and construction sites) will be prohibited from emitting or reducing emissions. Finally, these pollution reduction measures not only prevent serious pollution events from occurring, but may also lead to a decrease in pollutant concentration levels.

8. Lines 378-382, the reasons can be various such as the change of meteorological conditions, so please give your evidence here.

Response. We fully agree with the reviewer's opinion that there are many factors that may cause a decrease in pollutant concentration. Here, we infer that the reduction in pollutant concentration is closely related to the implementation of emission reduction policies. This can be supported by the changes in the composition of PM$_{2.5}$ sources before and after the release of the "orange" haze alarm.

Compared to before the "orange" haze alarm, many strict pollutant reduction measures were implemented in Chengdu and surrounding areas during the alarm period, such as the time limit for motor vehicles was tightened, all open-air operations and all kinds of construction were prohibited and industrial enterprises were instructed to cease production or had limitations imposed on their production depending on the type of goods they were producing. Correspondingly, the contributions of sources related to these activities, such as vehicular emission, coal combustion, industrial processes and dust, have decreased by 7.9%, 16.5%, 6.3% and 10.9%, respectively (Fig. R1). In fact, the haze alarm period in this study corresponds to more unfavorable meteorological conditions. According to our observations, from NP-2 to Haze-2 ("orange" haze alarm period), the relative humidity increases from 50±6% to 72±8%, while the wind speed decreases from 0.7±0.4 m s$^{-1}$ to 0.6±0.4 m s$^{-1}$. This is conducive to the generation and accumulation of pollutants. Therefore, based on the observed

meteorological conditions and changes in source composition, we can infer that the decrease in pollutant concentration here is closely related to various emission reduction measures during the "orange" haze alarm. Of course, if the reviewer still believes that our inference lacks sufficient evidence, we are also happy to delete it.

[Figure]

**Figure S1: Composition of PM2.5 sources during NP-2 and Haze-2 periods (%).**

9. Table 2, is that better to show a figure instead of the table here as the numbers may not be that important.

Response. According to the reviewer's suggestion, the table has been presented in the form of figures (Fig. 7 and S5).

10. Line 495, should we look at Fig. 1(a) or Fig. 1(b)?

Response. Thanks for this important comment. In fact, we also found that it is unreasonable to go back to the beginning of the main text to search for the corresponding figures while reading the latter half of it. This may have caused trouble for readers.

At the same time, we found that our original content in this section was only a description of the geographical regions (cities) that the air masses passed during different periods and lack relevance to the topic of the study. Therefore, we have rewritten this section. The rewritten content not only compares the composition of air masses during different periods, but also relates air masses to pollution levels. Accordingly, we defined the six types of air masses as "clean" or "polluted" air masses. Then we found that the contribution of "clean" air masses during the NP1 period was higher than that of the two haze periods, and the air masses during the haze periods all passed through key potential source areas of pollutants. This proves the significant impact of regional transmission on pollution in Chengdu

at different time periods (Line 417-426).

11. Line 698, how did you calculate these contributions? Please give the definitions of the contributions and the methods obtaining them.

Response. We are very sorry for this unclear discussion. we have added the calculation of local contribution and regional transmission in section 2.4.2. The calculation formulas for the relative contribution of local sources and regional transmission is as follows:

Regional transmission $(PM_{2.5}) = \frac{\text{Sensitivity scenario }(PM_{2.5})}{\text{Baseline scenario }(PM_{2.5})} \times 100\%$

Local sources $(PM_{2.5}) = 1 - \text{Regional transmission }(PM_{2.5})$

12. Line 698, again, the local sources here include only anthropogenic sources.

Response. As our response to comment 5, we have added necessary explanations in the main text to ensure that readers can more accurately understand our research results (Line 205-206), i.e., "It is worth noting that this study mainly focuses on the contribution of anthropogenic sources, without considering the contribution of natural emissions, such as biogenic sources."

13. From Fig. S2 in the supplementary information, the model results of $PM_{2.5}$ and observations are not in a good consistency. Therefore, I doubt about the validity of the results shown in Section 3.6. In addition, my feeling is that the WRF-Chem simulation is not very associated with the topic of the paper, is that possible to remove this part?

Response. Thanks for this important comment. (1) At present, we have optimized the emission inventory and parameterization scheme, and re-run the simulation. The correlation between the simulated and observed values of the new simulation has been improved to 0.73, the NMB has been reduced to -24.2%, and the NME has been reduced to 27.1% (Fig. R2), which is a further improvement of the new simulation results compared with the previous results. Moreover, the evaluated parameters of the model simulation in this paper are within the acceptable range as reported by Huang et al (2021). Therefore, the simulation results in this study can be used for the analysis of the causes of $PM_{2.5}$

pollution in Chengdu. (2) When analyzing the formation mechanism of pollution, we found that there were differences in the sources and mixing states of particulate matter between the two haze periods. At the same time, there were significant differences in the composition of air masses and potential source areas between the two haze periods. These pieces of evidence suggest that regional transmission may have different impacts on the two haze events. However, we lack quantitative results. Therefore, we quantitatively studied the contribution of regional transmission at different time periods using the WRF-Chem model, which is crucial for clarifying the formation mechanism of pollution.

[Figure]

**Figure R2: Temporal variation in the simulated and observed surface PM$_{2.5}$ concentration at the Chengdu station.**

References

Huang, L., Zhu, Y., Zhai, H., Xue, S., Zhu, T., Shao, Y., et all., 2021. Recommendations on benchmarks for numerical air quality model applications in China - Part 1: PM$_{2.5}$ and chemical species. Atmos. Chem. Phys. 21, 2725-2743, https://doi.org/10.5194/acp-21-2725-2021, 2021.

14. Line 755, it is always better to share the data to the public instead of giving this sentence.

Response. Thanks for this important comment. According to the reviewer's suggestion, we have shared the data (both main text and supplementary materials) with the public.

15 In addition, the language of this manuscript has been edited by a professional organization, and the language editing certificate is as follows:

**Certificate**

[Figure]

| Reference number: 2023-HuangXiaojuan-2-R1 | Date: 28 November 2023 |
|---|---|
| **Contact author:** Junke Zhang | **Manuscript:** Chemical composition, sources and formation mechanism of urban $PM_{2.5}$ in Southwest China: A case study in January 2023 |

This document certifies that the above-detailed manuscript was edited by a native English-speaking expert at LucidPapers on the date stated.

Following the editing process, the editor's overall assessment is that:

| | |
|---|---|
| The manuscript will be ready for consideration by the target journal once the edits have been checked and approved/rejected as necessary. | |
| **The manuscript may require modifications to the text in response to the editor's changes and comments/queries, but a second check of any such modifications is unlikely to be needed before sending to the target journal.** | ☑ |
| The manuscript requires modifications to the text in response to the editor's changes and comments/queries, and a second check of any such modifications might be advisable before sending to the target journal. | |
| The manuscript requires modifications to the text in response to the editor's changes and comments/queries, and a second check of these changes is recommended before sending to the target journal. | |
| The manuscript requires major changes, rewriting and restructuring and a second edit of the entire paper is strongly recommended before sending to the target journal. | |

Signed:

Colin Smith
Chief Editor
LucidPapers

Email: colin.smith@lucidpapers.com
Website: http://www.lucidpapers.com

---

## Author Comment (AC2)

Dear editor and reviewer,

Thank you very much for the comments and suggestions, which contribute to improve the quality of our paper. We have replied all comments and suggestions in our point-by-point response attached below. In order to highlight the changes what we have done, the color of the revised text will become blue.

**Response to Anonymous Referee #2**

**RC2.** In this study, the authors investigated the characteristics of aerosol chemical compositions by bulk-chemical and single-particle observations, analyzed the sources and formation mechanism of $PM_{2.5}$ pollution at the beginning of 2023 in Chengdu. The results of this observation experiment, including chemical components, meteorological conditions and source analysis, are comprehensive introduced in this paper. However, with the lack of the innovation in methods or ideas, this work did not provide significant new insights and scientific highlights. I think this paper is more a presentation of observational data rather than a scientific analysis as research article. In addition, the observation period is only about half a month, which does not represent the pollution level in southwest China as described in title. I also agree with previous reviewers of this article about the lack of analysis and discussion of new scientific findings. As a result, I would encourage the authors to carry out in-depth analysis and innovative results with major revision. But much further work is still required at this stage.

Response. We are very grateful to the reviewer for giving us the chance to make revisions. We have made significant revisions to the structure and content of the manuscript. These revisions can be summarized as follows: (1) We have simplified some content that lacks innovation (section 3.1.1) or placed it in supplementary materials (Table 1 and the introduction to PMF factors) to ensure that the main text presents more important scientific information. (2) We conducted a more in-depth analysis of the obtained results: (a) when analyzing the regional transmission of pollutants, we not only analyzed the air mass transmission during different pollution periods, but also added the analysis results of WCWT (Line 406-426), which can directly reflect the concentration contribution of regional transmission to the observation station; (b) we analyzed the sources and mixing structures of individual particles in the two periods, and provided more scientific information and evidence for analyzing their

formation mechanisms (section 3.3.4); (c) in order to provide readers with a clearer understanding of the evolution characteristics of pollution during the observation period, we have drawn a conceptual model of pollution evolution (Fig. 12), which displays the evolution characteristics of various meteorological and pollution indicators, particle mixing states and sources, and pollution formation mechanisms in the study period; (d) we discussed the significance of this study, particularly the importance of the TEM-EDS results in studying particle health and climate environmental effects (section 4.2). (3) We optimized the figures of the manuscript, such as presenting the content of Table 2 in the original manuscript in the form of figures and adding a conceptual model figure of pollution evolution. (4) Although our study period is short, it cannot be ignored that this period includes two typical haze processes. Clarifying their sources and formation mechanisms is also of great value for future air pollution control. Therefore, referring to similar studies in the past (Zhang et al., 2019; Zhu et al., 2016; Liu et al., 2013), we have defined our study as a "case study". Correspondingly, the title has been modified to:"**Chemical composition, sources and formation mechanism of urban PM$_{2.5}$ in Southwest China: A case study in January 2023**". We believe that through this major revision, the manuscript can present more and in-depth scientific information, allowing readers to better understand the sources and formation mechanisms of haze pollution in Chengdu.

Reference:

Liu, X. G., Li, J., Qu, Y., Han, T., Hou, L., Gu, J., Chen, C., Yang, Y., Liu, X., Yang, T., Zhang, Y., Tian, H., and Hu, M.: Formation and evolution mechanism of regional haze: a case study in the megacity Beijing, China, Atmos. Chem. Phys., 13, 4501-4514, https://doi.org/10.5194/acp-13-4501-2013, 2013.

Zhang, W., Zhang, Y. L., Cao, F., Xiang, Y., Zhang, Y., Bao, M., Liu, X., and Lin, Y. C.: High time-resolved measurement of stable carbon isotope composition in water-soluble organic aerosols: method optimization and a case study during winter haze in eastern China, Atmos. Chem. Phys., 19, 11071-11087, https://doi.org/10.5194/acp-19-11071-2019, 2019.

Zhu, X. W., Tang, G. Q., Hu, B., Wang, L. L., Xin, J. Y., Zhang, J. K., Liu, Z. R., Münkel, C., and S., W. Y.: Regional pollution and its formation mechanism over North China Plain: A case study with ceilometer observations and model simulations, J. Geophys. Res-Atmos., 121, D14574,

https://doi.org/10.1002/2016JD025730, 2016.

1. L171-174: The diagram in the upper left corner of Figure 1(a) needs to remove the information of road network.

Response. Thanks for this important comment. In fact, we also feel that the expression of this diagram is not clear enough. Currently, we have provided a clearer map that presents the terrain characteristics of the study area and the detailed location of the observation station (Fig. 1), which is beneficial for readers to have a clearer understanding of the research area. Meanwhile, the analysis of air mass transmission is an auxiliary content, and relevant figure have been placed in the supplementary materials (Fig. S3).

2. L217: Please add references about density of 2 g cm$^{-3}$.

Response. The necessary references has been added (Line 164-165), such as Xu et al. (2021), Li et al. (2021), Li and Shao (2009) and Marple et al. (1993).

References:

Li, W. and Shao, L.: Transmission electron microscopy study of aerosol particles from the brown hazes in northern China, J. Geophys. Res., 114, D09302, https://doi.org/10.1029/2008jd011285, 2009.

Li, W., Teng, X., Chen, X., Liu, L., Xu, L., Zhang, J., Wang, Y., Zhang, Y., and Shi, Z.: Organic coating reduces hygroscopic growth of phase-separated aerosol particles, Environ. Sci. Technol., 55, 16339-16346, https://doi.org/10.1021/acs.est.1c05901, 2021.

Marple, V. A., Rubow, K. L., and Olson, B. A.: Inertial, gravitational, centrifugal, and thermal collection techniques, in aerosol measurement, Aerosol Meas., 8, 206–233, https://doi.org/10.1002/9781118001684.ch8, 1993.

Xu, L., Liu, X., Gao, H., Yao, X., Zhang, D., Bi, L., Liu, L., Zhang, J., Zhang, Y., Wang, Y., Yuan, Q., and Li, W.: Long-range transport of anthropogenic air pollutants into the marine air: insight into fine particle transport and chloride depletion on sea salts, Atmos. Chem. Phys., 21, 17715–17726,

https://doi.org/10.5194/acp-21-17715-2021, 2021.

3. L372: Add the definition of "orange" haze alarm.

Response. Due to the haze alarm in Chengdu includes three levels, namely "yellow", "orange" and "red" alarms. And their definitions are relatively complex. Therefore, in order to provide readers with a comprehensive understanding of haze alarm information, we have introduced their definitions in the supplementary materials (Text S2).

4. L495-505: This description seems superfluous and does not fit the main idea of the article.

Response. Thanks for this important comment. We also found that our original content in this section was only a description of the geographical regions (cities) that the air masses passed and lack relevance to the topic of the study. Therefore, we have rewritten this section (Line 417-426). The rewritten content not only compares the composition of air masses during the three periods, but also relates air masses to pollution levels. Accordingly, we defined the six types of air masses as "clean" or "polluted" air masses. Then we found that the contribution of "clean" air masses during the NP1 period was higher than that of the two haze periods, and the air masses during the haze periods all passed through key potential source areas of pollutants. This proves the significant impact of regional transmission on pollution in different periods.

5. L556-558: The particulate removal process is complex and there is no direct indication that this is influenced by the hygroscopicity here.

Response. Thanks for this important comment. Our analysis approach in this section is as follows: (1) From Haze-1 to NP-2, the $PM_{2.5}$ mass concentration decreased by 85 μg m$^{-3}$ within 31 hours. Correspondingly, this process is accompanied by short-term precipitation and strong winds, which are considered meteorological factors that cause a rapid decrease in pollutant concentration in the short term (Tsai et al., 2014; Zhang et al., 2015; Hu et al., 2021). (2) During the rapid decrease in $PM_{2.5}$ concentration, the contribution of secondary inorganic species (SNA measured by filter sampling method) or SIA-containing particles (OM-SIA and OM-SIA-soot particles measured by TEM-EDS method) with strong hygroscopicity decreased by 15.6% and 27.1%, respectivley, while the

contribution of hydrophobic carbon species (OM and EC measured by filter sampling method) or particles (OM, soot and OM-soot measured by TEM-EDS method) increased by 2.8% and 23.7%, respectivley. Therefore, we infer that the significant decrease in $PM_{2.5}$ concentration and changes in its chemical composition are closely related to the precipitation and wind. Of course, if the reviewer still believes that our inference lacks sufficient evidence, we are also very happy to delete it.

Reference:

Hu, W., Zhao, T., Bai, Y., Kong, S., Xiong, J., Sun, X., Yang, Q., Gu, Y., and Lu, H.: Importance of regional $PM_{2.5}$ transport and precipitation washout in heavy air pollution in the Twain-Hu Basin over Central China: Observational analysis and WRF-Chem simulation, Sci. Total Environ., 758, 143710, https://doi.org/10.1016/j.scitotenv.2020.143710, 2021.

Tsai, Y. I., Kuo, S. C., Young, L. H., Hsieh, L. Y., and Chen, P. T.: Atmospheric dry plus wet deposition and wet-only deposition of dicarboxylic acids and inorganic compounds in a coastal suburban environment, Atmos. Environ., 89, 696-706, https://doi.org/10.1016/j.atmosenv.2014.03.013, 2014.

Zhang, Z., Zhang, X., Gong, D., Quan, W., Zhao, X., Ma, Z., and Kim, S. J.: Evolution of surface $O_3$ and $PM_{2.5}$ concentrations and their relationships with meteorological conditions over the last decade in Beijing, Atmos. Environ., 108, 67-75, https://doi.org/10.1016/j.atmosenv.2015.02.071, 2015.

6. L605-620: Figure 7 and its associated descriptions are best moved to the supplement file.

Response. According to the reviewer's comment, we have moved Figure 7 and its associated descriptions to the supplementary materials (Text S3).

7. L698-704: How were the contributions of local sources and regional transmission calculated?

Response. We are very sorry for this unclear discussion. we have added the calculation of local contribution and regional transmission in section 2.4.2, The calculation formulas for the relative contribution of local sources and regional transmission is as follows:

Regional transmission $(PM_{2.5})=\frac{\text{Sensitivity scenario }(PM_{2.5})}{\text{Baseline scenario }(PM_{2.5})}\times100\%$

Local sources $(PM_{2.5})=1-$Regional transmission $(PM_{2.5})$

8. L633: Please explain what is meant by "non-exhaust emissions".

Response. Thanks for this important comment. A detailed introduction to "non-exhaust emissions" can be found in Charron et al. (2019). However, we found that this discussion is not closely related to the topic of this study, and therefore it has been deleted.

Reference:

Charron, A., Polo-Rehn, L., Besombes, J.-L., Golly, B., Buisson, C., Chanut, H., Marchand, N., Guillaud, G., and Jaffrezo, J.-L.: Identification and quantification of particulate tracers of exhaust and non-exhaust vehicle emissions, Atmos. Chem. Phys., 19, 5187-5207, https://doi.org/10.5194/acp-19-5187-2019, 2019.

9. The grammar of the essay needs a thorough examination.( for example, L125: "investigate" ; L257: "sensitivity"; L315: "mitigate"…)

Response. Thanks for this important comment. The language of this manuscript has been edited by a professional organization, and the language editing certificate is as follows:

**Certificate**

[Figure]

| | |
|---|---|
| **Reference number**: 2023-HuangXiaojuan-2-R1 | **Date**: 28 November 2023 |
| **Contact author:** Junke Zhang | **Manuscript:** Chemical composition, sources and formation mechanism of urban $PM_{2.5}$ in Southwest China: A case study in January 2023 |

This document certifies that the above-detailed manuscript was edited by a native English-speaking expert at LucidPapers on the date stated.

Following the editing process, the editor's overall assessment is that:

| | |
|---|---|
| The manuscript will be ready for consideration by the target journal once the edits have been checked and approved/rejected as necessary. | |
| **The manuscript may require modifications to the text in response to the editor's changes and comments/queries, but a second check of any such modifications is unlikely to be needed before sending to the target journal.** | ☑ |
| The manuscript requires modifications to the text in response to the editor's changes and comments/queries, and a second check of any such modifications might be advisable before sending to the target journal. | |
| The manuscript requires modifications to the text in response to the editor's changes and comments/queries, and a second check of these changes is recommended before sending to the target journal. | |
| The manuscript requires major changes, rewriting and restructuring and a second edit of the entire paper is strongly recommended before sending to the target journal. | |

Signed:

Colin Smith
Chief Editor
LucidPapers

Email: colin.smith@lucidpapers.com
Website: http://www.lucidpapers.com

---

## Author Response (AR2)

Dear editor and reviewers,

Thank you very much for the comments and suggestions, which contribute to improve the quality of our paper. We have replied all comments and suggestions in our point-by-point response attached below. In order to highlight the changes what we have done, the color of the revised text will become blue.

**Response to Anonymous referee #3**

Authors have carefully addressed the comments raised by two reviewers. I respect author's efforts. In its present form, the work is acceptable for final publication in ACP.

Response. We greatly appreciate the reviewer's review and recognition of this manuscript.

**Response to Anonymous referee #4**

The Sichuan Basin is one of the regions severely affected by haze in China. This study focused on Chengdu, the capital of Sichuan Province, and conducted in-depth analysis of the haze processes that occurred in early 2023 through field observations and model. By reviewing the review process of this manuscript, I found that the quality of the revised manuscript has significantly improved compared to the original version. Meanwhile, the results obtained in this study are interesting and can further enhance our understanding of the formation mechanism of air pollution in the city. However, there are still some scientific issues that need to be clarified or improved in the current version. Therefore, a revision is necessary before considering acceptance.

1. Line 50-55, some references about haze and non-haze study as well as control policy evaluation should be cited. Such as: Variations in $PM_{2.5}$, TSP, BC, and trace gases ($NO_2$, $SO_2$, and $O_3$) between haze and non-haze episodes in winter over Xi'an, China. Atmospheric Environment, 2015, 112, 64-71. Saccharides in summer and winter $PM_{2.5}$ over Xi'an, Northwestern China: Sources, and yearly variations of biomass burning contribution to $PM_{2.5}$. Atmospheric Research, 2018, 214, 410-417.

Inter-annual variability of wintertime PM$_{2.5}$ chemical composition in Xi'an, China: Evidences of changing source emissions. Sci. Total Environ. 545, 546–555.

Response. Thanks for providing these valuable references and they have already been cited in the manuscript (Line 38 and 51).

2. Line 60-62. I suggest the authors provide more comprehensive evidence and data on the improvement of air quality in China in recent years. It is insufficient to provide an example based solely on the study results from Beijing. In fact, there are already many comprehensive reports or literature on the implementation effectiveness of emission reduction policies.

Response. Thanks for this important comment. We have added a discussion on the improvement of air quality throughout China and the three major polluted areas in recent years (Line 58-62): "*For example, the annual mean PM$_{2.5}$ mass concentration in the China and its three key polluted areas, namely the Beijing-Tianjin-Hebei (BTH) and its surrounding areas, YRD and FWP, has decreased from 39, 60, 44 and 58 μg m$^{-3}$ in 2018 to 29, 44, 31 and 46 μg m$^{-3}$ in 2022, respectively. Meanwhile, the concentration of SO$_2$, one of the important gaseous precursors of PM$_{2.5}$, has decreased from 14, 20, 11 and 24 μg m$^{-3}$ to 9, 10, 7 and 9 μg m$^{-3}$, respectively (https://www.mee.gov.cn/hjzl/sthjzk/zghjzkgb/, last access: 10 January 2024)".* These results can better support our viewpoint.

3. Line 100-102. "Despite numerous studies having used multiple methods to investigate the physical……" Please provide necessary references.

Response. The necessary references have been added, such as Chen et al., 2022, Huang et al., 2018, Huang et al., 2021a, Tao et al., 2014 and Wang et al., 2018a (Line 103-104).

References:

Chen, L., Zhang, J., Huang, X., Li, H., Dong, G., and Wei, S.: Characteristics and pollution formation mechanism of atmospheric fine particles in the megacity of Chengdu, China, Atmos. Res., 273, 106172, https://doi.org/10.1016/j.atmosres.2022.106172, 2022.

Huang, X., Zhang, J., Zhang, W., Tang, G., and Wang, Y.: Atmospheric ammonia and its effect on PM$_{2.5}$ pollution in urban Chengdu, Sichuan Basin, China, Environ. Pollut., 291, 118195, https://doi.org/10.1016/j.envpol.2021.118195, 2021a.

Huang, X., Zhang, J., Luo, B., Wang, L., Tang, G., Liu, Z., Song, H., Zhang, W., Yuan, L., and Wang, Y.:

Water-soluble ions in $PM_{2.5}$ during spring haze and dust periods in Chengdu, China: Variations, nitrate formation and potential source areas, Environ. Pollut., 243, 1740-1749, https://doi.org/10.1016/j.envpol.2018.09.126, 2018.

60    Tao, J., Gao, J., Zhang, L., Zhang, R., Che, H., Zhang, Z., Lin, Z., Jing, J., Cao, J., and Hsu, S. C.: $PM_{2.5}$ pollution in a megacity of southwest China: source apportionment and implication, Atmos. Chem. Phys., 14, 8679-8699, https://doi.org/10.5194/acp-14-8679-2014, 2014.

Wang, H., Tian, M., Chen, Y., Shi, G., Liu, Y., Yang, F., Zhang, L., Deng, L., Yu, J., Peng, C., and Cao, X.: Seasonal characteristics, formation mechanisms and source origins of $PM_{2.5}$ in two megacities in Sichuan Basin, China, Atmos. Chem. Phys., 18, 865-881,

65    https://doi.org/10.5194/acp-18-865-2018, 2018a.

4. According to section 3.1, a haze event usually lasts for several days, while the collection time for each single particle sample was only 30 s to 3 min. I want to know how the authors ensure that the single particles they analyze are representative.

70    Response. We fully understand the concerns of the reviewer. However, TEM is a research method that analyzes the collected particles one by one. Therefore, it requires that the particles collected onto copper TEM grids should not overlap and each particle should be clearly visible. As is well known, the number concentration of particulate matter in the atmosphere is usually at a very high level, especially during pollution periods. This determines that the collection time for each sample cannot be too long.

75    In fact, previous studies on air pollution using TEM method have adopted the same research approach as our study, and the obtained study results are considered reliable (Li and Shao, 2009; Li et al., 2014; Li et al., 2015; Xu et al., 2020; Xu et al., 2021; Zhang et al., 2021b).

References

Li, W. and Shao, L.: Transmission electron microscopy study of aerosol particles from the brown hazes

80    in northern China, J. Geophys. Res-Atmos., 114, D09302, https://doi.org/10.1029/2008jd011285, 2009.

Li, W., Shao, L., Shi, Z., Chen, J., Yang, L., Yuan, Q., Yan, C., Zhang, X., Wang, Y., Sun, J., Zhang, Y., Shen, X., Wang, Z., and Wang, W.: Mixing state and hygroscopicity of dust and haze particles before leaving Asian continent, J. Geophys. Res-Atmos., 119, 1044-1059,

85    https://doi.org/10.1002/2013jd021003, 2014.

Li, W. J., Chen, S. R., Xu, Y. S., Guo, X. C., Sun, Y. L., Yang, X. Y., Wang, Z. F., Zhao, X. D., Chen, J. M., and Wang, W. X.: Mixing state and sources of submicron regional background aerosols in the northern Qinghai-Tibet Plateau and the influence of biomass burning, Atmos. Chem. Phys., 15, 13365-13376, https://doi.org/10.5194/acp-15-13365-2015, 2015.

90  Xu, L., Fukushima, S., Sobanska, S., Murata, K., Naganuma, A., Liu, L., Wang, Y., Niu, H., Shi, Z., Kojima, T., Zhang, D., and Li, W.: Tracing the evolution of morphology and mixing state of soot particles along with the movement of an Asian dust storm, Atmos. Chem. Phys., 20, 14321-14332, https://doi.org/10.5194/acp-20-14321-2020, 2020.

Xu, L., Liu, X., Gao, H., Yao, X., Zhang, D., Bi, L., Liu, L., Zhang, J., Zhang, Y., Wang, Y., Yuan, Q.,
95  and Li, W.: Long-range transport of anthropogenic air pollutants into the marine air: insight into fine particle transport and chloride depletion on sea salts, Atmos. Chem. Phys., 21, 17715–17726, https://doi.org/10.5194/acp-21-17715-2021, 2021.

Zhang, J., Yuan, Q., Liu, L., Wang, Y., Zhang, Y., Xu, L., Pang, Y., Zhu, Y., Niu, H., Shao, L., Yang, S., Liu, H., Pan, X., Shi, Z., Hu, M., Fu, P., and Li, W.: Trans-regional transport of haze particles from
100  the North China Plain to Yangtze River Delta during Winter, J. Geophys. Res-Atmos., 126, D033778, https://doi.org/10.1029/2020jd033778, 2021b.

5. Line 263-264. "In particular, at the beginning of 2023, Chengdu has become the city with the highest number of motor vehicles in China" Please provide the data source.

105  Response. In fact, the first comment on the number of motor vehicles in Chengdu appeared in the introduction section, and we provided necessary references, i.e. https://www.mps.gov.cn/n2254098/n4904352/c9244719/content.html (Line 94).

6. Line 294-295. "although there was a significant increase in pollution at night, the pollution level in
110  the daytime was lower than that on 3 February." Why? If the emission reduction policies during the haze alarm period have had a positive effect, what are the reasons for the high values at night? What are the main sources of pollution?

Response. Thanks for this important comment. (1) The emission reduction policies during the haze alarm period mainly target various anthropogenic sources, such as mobile sources, industrial processes
115  and construction, which almost all appear during the daytime. This is why after the implementation of

emission reduction policies, the reduction of pollutants mainly occurred during the daytime. (2) The reasons for the high-level pollution at night are complex, such as a shallow boundary layer, secondary generation of pollutants or regional transmission. In fact, the phenomenon that the reviewer is concerned about occurred during the Haze-2 period, and according to our analysis, this pollution process is mainly caused by the regional transmission of secondary pollutants.

7. Line 369. "S-rich particles"? Is this a new type of particulate matter? If so, please provide necessary introduction.

Response. The "S-rich particles" should be "SIA particle" and it has been corrected (Line 347). We are very sorry for our carelessness.

8. Line 535-539. "This is mainly related to the various pollution reduction measures implemented in Chengdu, and even the broader SCB, in the past decade, such as……" I think some references are necessary here.

Response. The necessary reference has been added (Line 508-509), i.e., http://sthjt.sc.gov.cn/sthjt/c106120/2018/12/28/8f2a9dead56c4605ad2d26fe2e2fac43.shtml.

9. Line 577-579. "This means that if Chengdu's air quality is to achieve further improvement after reaching the current CNAAQS, it may need more attention paid to the contribution of local sources". I think this statement is unreasonable. In fact, the contribution of regional transmission reached 46%. Such a high contribution cannot be ignored in the future pollution control process. Therefore, I suggest that the authors revise it.

Response. Thanks for this important comment. We fully agree with the reviewer's viewpoint and the unreasonable statement has been removed. According to our analysis results, the future control of air pollution in Chengdu requires synchronous emission reduction of local sources and regional transmission.

10. Although the chemical composition results obtained based on bulk-chemical and single-particle analysis exhibited strong consistency, there is a significant difference in their magnitude of change. For example, from Haze-1 to NP-2, the contributions of SNA and SIA particles decreased by 15.6% and

27.1%, respectively; from NP-2 to Haze-2, the contributions of SNA and SIA particles increased by 24.2% and 42.4%, respectively. What factors are causing this difference?

Response. Thanks for this important comment. This is because SNA and SIA are two types of data, that is, SNA is the **mass** concentration of secondary inorganic components, while SIA is the **number** concentration of secondary inorganic particles. We must note that: (1) due to the differences in particle size distribution among different species, their contributions in number concentration and mass concentration are usually different. For example, in our study, the mass contribution of EC was only 5.8%–6.9%, while the corresponding species, namely soot-related particles, had a number contribution of 12.5%–23.9%; (2) the individual particles measured by TEM are usually a mixture of particles from multiple chemical components. These two factors ultimately lead to different degrees of variation in the pollution evolution for the two types of data. In fact, similar phenomena have also appeared in previous studies, such as (Zhang et al., 2020b; Zhang et al., 2021a). For example, in the study of Zhang et al. (2020b), they found that from the heavy Haze-I to Haze-II events, secondary inorganic ions (mass) significantly decreased from 62%-66% of the total $PM_{2.5}$ mass to 31%–35%, but OM (mass) markedly increased from 27%-30% to 53%-60%. Meanwhile, the S-OM particles fractions (number) significantly decreased from 60%-74% to 30%-32%, but K-OM fractions (number) largely increased from 4%-5% to 50 %-52%.

References:

Zhang, J., Liu, L., Xu, L., Lin, Q., Zhao, H., Wang, Z., Guo, S., Hu, M., Liu, D., Shi, Z., Huang, D., and Li, W.: Exploring wintertime regional haze in northeast China: role of coal and biomass burning, Atmos. Chem. Phys., 20, 5355-5372, https://doi.org/10.5194/acp-20-5355-2020, 2020b.

Zhang, J., Huang, X., Yu, Y., Liu, Q., Zhang, J., Song, H., and Wang, Y.: Insights into the characteristics of aerosols using an integrated single particle–bulk chemical approach, Atmos. Res., 250, 105374, https://doi.org/10.1016/j.atmosres.2020.105374, 2021a.

11. Line 626-628. "……with the reduction of pollution and the enhancement of atmospheric oxidation in recent years (Zhao et al., 2020; Wang et al., 2023), the formation mechanism of haze is also undergoing dynamic changes……" What changes in pollution mechanisms have been found in this study compared to previous results? I think some necessary discussions need to be added to the main text.

Response. Thanks for this valuable comment. According to the reviewer's suggestion, we compared our study results with the pollution formation mechanism reported in other regions of China (Northeast and North China Plain) and Chengdu in previous winters. Then emphasized the new results obtained in our study (Line 538-547): "*We found that the formation mechanism of haze in this study is different*

180 *from previous winter study results in other regions in China, such as northern China. For example, Zhang et al. (2020b) found that residential coal burning and biomass burning were important factors causing winter haze in Northeast China. While, the contribution of industrial emissions to the formation of winter haze in the NCP region was much higher than that in Northeast China (Ma et al., 2016). Meanwhile, compared to previous winter studies in Chengdu (Liao et al., 2017; Li et al., 2017;*

185 *Tao et al., 2013), the haze formation in this study presented some new characteristics. For example, (1) the key potential source areas during the haze period have shifted from the southeast in 2013 to the south; (2) mobile sources played a more important role, while the contributions of biomass burning and dust sources were significantly weaker; (3) the contribution of nitrate to the formation of heavy pollution was more prominent. This means that in order to develop efficient pollution reduction*

190 *policies, it is very necessary to conduct targeted and timely research on the characteristics, sources and formation mechanisms of haze in the areas of concern.*"

12. In section 3.4 source apportionments, the sources profiles of each factors should be given, and related references should be cited to support the source apportionment results.

195 Response. Thanks for this important comment. In fact, an introduction to the sources profiles of each factor was included in the main text of the original manuscript. However, during the first round of review process, one referee suggested that these materials should be included in the supplementary materials. We also believe that this introduction is auxiliary and not the core content of our manuscript. Therefore, the relevant content has been included in the supplementary materials now (S3). We hope to

200 gain the understanding of the reviewer.

13. Figures: (1) The marking of "fly ash/metal particles" in Figure 9 is not clear enough, please modify it. (2) Figure 11 is not clear enough, and its quality needs to be improved. (3) Figure 12. It is "relative humidity" rather than "humidity".

205 Response. We have completed the corresponding revisions. At the same time, we checked the quality

of all figures to ensure that they can be easily understood.

---

## Author Response (AR3)

Dear editor and reviewer,

Thank you very much for the comments and suggestions, which contribute to improve the quality of our paper. We have replied all comments and suggestions in our point-by-point response attached below. In order to highlight the changes what we have done, the color of the revised text will become blue.

Dear Authors, I greatly appreciate the effort you made to improve the manuscript. Please consider the following additional comments:

Response to Comment 5, from referee #4: "In fact, the first comment on the number of motor vehicles in Chengdu appeared in the introduction section, and we provided necessary references, i.e. https://www.mps.gov.cn/n2254098/n4904352/c9244719/content.html (Line 94)".
The given reference is not in English. Please provide a reference that is understandable to all readers of ACP.
Response. Thanks for this important comment. We fully agree with that English reference is crucial for readers to understand scientific research results. However, the content here is about the ranking of Chengdu's motor vehicle ownership in China, and the latest data was released in October 2023, which is only three months away from now. Therefore, there is currently no reference reporting on the relevant data. Therefore, we have to cite relevant reports from government departments. However, according to this important comment, we have replaced other non English references. For example, "*such as the "Atmospheric Pollution Prevention and Control Action Plan (APPCAP)" during 2013–2017 (https://www.gov.cn/zwgk/2013-09/12/content_2486773.htm, last access: 10 January 2024) and the "Three-year Action Plan to Win the Blue Sky Defense War (BSDW)" during 2018–2020 (https://www.gov.cn/zhengce/content/2018-07/03/content_5303158.htm, last access: 10 January 2024)*" has been revised as "*such as the "Atmospheric Pollution Prevention and Control Action Plan (APPCAP)" during 2013–2017 and the "Three-year Action Plan to Win the Blue Sky Defense War (BSDW)" during 2018–2020 (Geng et al., 2019; Huang et al., 2014; Huang et al., 2021b; Wang et al., 2022a)*" (Line 53-55).

Comments 6 and 10, from referee #4.
Please include the response to these comments in the text in order to clarify the issue for other readers.
Response. Thanks for this important comment. The responses have been appropriately added to the main text:
Line 278-280: "*Due to these emission reduction measures mainly targeting pollution sources that occur during the daytime. Therefore, although there was a significant increase in pollution at night, the pollution level in the daytime in the following days was lower than that on 3 February.*"
Line 455-458: "*It can be seen that although single-particle and bulk-chemical methods analyzed the chemical composition of particulate matter from two different perspectives, and there were differences in the contribution changes of similar species (such as SIA particles and SNA) with pollution evolution, but the two types of data exhibited similar trends. By combining these results, we can have a clearer understanding of the formation mechanism of pollution.*"

Comment 12, from referee #4: "In section 3.4 source apportionments, the sources profiles of each factor

should be given, and related references should be cited to support the source apportionment results."

I agree that the source profiles can be described in the supplemental material document. However, the references to support the association of a source to each profile are still missing.

Response. Thanks for this important comment. We have added all necessary references in the supplementary materials.

In addition, just after submitting my decision regarding the previous version of the manuscript, I received an additional report from an anonymous referee, which I am copying below. Although their comments are minor, I think that they can further improve the manuscript. Hence, I ask you to consider them too.

\*\*\*\*\*\*\*\*\*\*\*\*\*\*\*\*\*\*\*\*\*\*\*\*\*\*\*\*\*\*\*\*\*\*\*\*\*\*\*\*\*\*\*\*\*\*\*\*\*\*\*\*\*\*\*\*\*\*\*\*\*\*\*\*\*\*\*\*\*\*\*\*\*\*\*\*\*\*\*\*

**Response to Anonymous referee**

In this study, the authors synthesized multiple methods to study the characteristics, sources and formation mechanisms of atmospheric $PM_{2.5}$ pollution in Chengdu. They captured two typical pollution processes and conducted in-depth analysis of their formation mechanisms, and the results can be used as a reference for local pollution reduction. Compared to previous studies on this city, the research methods and results in this study have a certain degree of innovation. In addition, after carefully reading the first round of review comments and the authors' responses, I believe that the authors responded to the comments with high quality and made necessary modifications, which significantly improved the quality of the manuscript. However, there are still some issues that need to be further modified and improved before being considered for acceptance. The detailed comments are as follows.

1. The authors conducted a detailed analysis of the formation mechanism of two haze processes during the observation period. I would like to know if there is any difference in the formation mechanism of winter haze in Chengdu compared to previous studies? Therefore, it is very necessary to compare with previous study results and add relevant discussions.

Response. Thanks for this valuable comment. In fact, the referee #4 has raised similar comment, and we have added corresponding discussions. We compared our study results with the pollution formation mechanism reported in other regions of China (Northeast and North China Plain) and Chengdu in previous winters. Then emphasized the new results obtained in our study (Line 547-556): "*We found that the formation mechanism of haze in this study is different from previous winter study results in other regions in China, such as northern China. For example, Zhang et al. (2020b) found that residential coal burning and biomass burning were important factors causing winter haze in Northeast China. While, the contribution of industrial emissions to the formation of winter haze in the NCP region was much higher than that in Northeast China (Ma et al., 2016). Meanwhile, compared to previous winter studies in Chengdu (Liao et al., 2017; Li et al., 2017; Tao et al., 2013), the haze formation in this study presented some new characteristics. For example, (1) the key potential source areas during the haze period have shifted from the southeast in 2013 to the south; (2) mobile sources played a more important role, while the contributions of biomass burning and dust sources were significantly weaker; (3) the contribution of nitrate to the formation of heavy pollution was more prominent. This means that in order to develop efficient pollution reduction policies, it is very necessary to conduct targeted and timely research on the characteristics, sources and formation mechanisms of haze in the areas of concern.*"

Reference.

Li, L. L., Tan, Q. W., Zhang, Y., Feng, M., Qu, Y., An, J. L., and Liu, X. A.: Characteristics and source apportionment of PM$_{2.5}$ during persistent extreme haze events in Chengdu, southwest China, Environ. Pollut., 230, 718-729, https://doi.org/10.1016/j.envpol.2017.07.029, 2017.

Liao, T. T., Wang, S., Ai, J., Gui, K., Duan, B. L., Zhao, Q., Zhang, X., Jiang, W. T., and Sun, Y.: Heavy pollution episodes, transport pathways and potential sources of PM$_{2.5}$ during the winter of 2013 in Chengdu (China), Sci. Total Environ., 584-585, 1056-1065, https://doi.org/10.1016/j.scitotenv.2017.01.160, 2017.

Ma, L., Li, M., Zhang, H. F., Li, L., Huang, Z. X., Gao, W., Chen, D. H., Fu, Z., Nian, H. Q., Zou, L. L., Gao, J., Chai, F. H., and Zhou, Z.: Comparative analysis of chemical composition and sources of aerosol particles in urban Beijing during clear, hazy, and dusty days using single particle aerosol mass spectrometry, J. Clean. Prod., 112, 1319-1329, https://doi.org/10.1016/j.jclepro.2015.04.054, 2016.

Tao, J., Zhang, L. M., Engling, G., Zhang, R. J., Yang, Y. H., Cao, J. J., Zhu, C. S., Wang, Q. Y., and Luo, L.: Chemical composition of PM$_{2.5}$ in an urban environment in Chengdu, China: Importance of springtime dust storms and biomass burning, Atmos. Res., 122, 270-283, https://doi.org/10.1016/j.atmosres.2012.11.004, 2013.

Zhang, J., Liu, L., Xu, L., Lin, Q. H., Zhao, H. J., Wang, Z. B., Guo, S., Hu, M., Liu, D. T., Shi, Z. B., Huang, D., and Li, W. J.: Exploring wintertime regional haze in northeast China: role of coal and biomass burning, Atmos. Chem. Phys., 20, 5355-5372, https://doi.org/10.5194/acp-20-5355-2020, 2020b.

2. Line 107-108. The authors mentioned that "At the beginning of 2023, Chengdu experienced several severe haze events, during which the observed PM$_{2.5}$ mass concentration frequently exceeded the CNAAQS." Please provide more quantitative descriptions about the severity of this pollution, which will help readers understand the necessity of this study.

Response. According to the reviewer's suggestion, we have added a description of the pollution situation at the beginning of 2023 (Line 108-110): "*At the beginning of 2023, Chengdu experienced several severe haze events. The longest pollution process lasted for 12 days, and the highest daily average of PM$_{2.5}$ mass concentration reached 156 μg m$^{-3}$, which is 2.1 times the CNAAQS (daily average of 75 μg m$^{-3}$). Meanwhile, the proportion of mild and more severe pollution days in January and February reached 37.3%.*"

3. Line 172. The numbers of single particles analyzed during the four periods were 1325, 1159, 995 and 1870, respectively. I want to know if these analyzed particles are representative? Especially in the third period, the number of particles is less than 1000. This is crucial for the feasibility of the study results.

Response. Thanks for this important comment. We fully agree with the reviewer's comment, i.e., a larger sample size helps to obtain more accurate analysis results. However, during the TEM-EDS analysis process, each individual particle needs to be analyzed one by one, which is a very time-consuming, labor-intensive and costly process. Therefore, many previous studies were based on the analysis of hundreds or thousands of individual particle measurements. For example, Li et al., (2009) analyzed 810 particles in Beijing; Xu et al., (2020) analyzed 412, 486, and 887 aerosol particles at an inland urban site and a coastal urban site in China and a coastal site in southwestern Japan, respectively; Fu et al., (2012)

analyzed 834 particles in Shanghai; Li et al., (2021) analyzed 310, 280, 292 particles in Beijing, Hangzhou and Lesser Khingan Mountains, respectively. These results are believed to accurately reflect the characteristics of air pollution. Based on these previous studies, the numbers of particles analyzed for our four periods were 1325, 1159, 995 and 1870, respectively. We believe that our research results are reliable. In addition, accroding to the reviewer's comment, we will analyze as many particles as possible in future research.

Reference.

Fu, H., Zhang, M., Li, W., Chen, J., Wang, L., Quan, X. and Wang, W.: Morphology, composition and mixing state of individual carbonaceous aerosol in urban Shanghai, Atmos. Chem. Phys., 12, 693-707, https://doi.org/10.5194/acp-12-693-2012, 2012.

Li, W. J. and Shao, L. Y.: Transmission electron microscopy study of aerosol particles from the brown hazes in northern China, J. Geophys. Res. Atmos., 114, D09302, https://doi.org/10.1029/2008jd011285, 2009.

Li, W. J., Teng, X. M., Chen, X. Y., Liu, L., Xu, L., Zhang, J., Wang, Y. Y., Zhang, Y. and Shi, Z. B.: Organic coating reduces hygroscopic growth of phase-separated aerosol particles, Environ. Sci. Technol, 55, 16339-16346, https://doi.org/10.1021/acs.est.1c05901, 2021.

Xu, L., Fukushima, S., Sobanska, S., Murata, K., Naganuma, A., Liu, L., Wang, Y. Y., Niu, H. Y., Shi, Z. B., Kojima, T., Zhang, D. Z. and Li, W. J.: Tracing the evolution of morphology and mixing state of soot particles along with the movement of an Asian dust storm, Atmos. Chem. Phy., 20, 14321-14332, https://doi.org/10.5194/acp-20-14321-2020, 2020.

4. Line 252-253. "This inevitably leads to a the difference in the formation mechanism of air pollution in these two regions in China (Wang et al., 2021)" Why?

Response. Thanks for this important comment. This is mainly because temperature and relative humidity directly affect the homogeneous and heterogeneous generation processes of $PM_{2.5}$ chemical components. The necessary explanations have been added (Line 236-239): "*Due to the direct influence of T and RH on the homogeneous and heterogeneous generation processes of $PM_{2.5}$ chemical components, such as nitrates, sulfates and secondary organic compounds (Wang et al., 2021; An et al., 2019). Therefore, this difference inevitably leads to differences in the formation mechanisms of air pollution between the north and south of China.*"

Reference.

An, Z., Huang, R. J., Zhang, R., Tie, X., Li, G., Cao, J., Zhou, W., Shi, Z., Han, Y., Gu, Z., and Ji, Y.: Severe haze in northern China: a synergy of anthropogenic emissions and atmospheric processes, Proc. Natl. Acad. Sci. U.S.A., 116, 8657-8666, https://doi.org/10.1073/pnas.1900125116, 2019.

Wang, Y., Hu, M., Hu, W., Zheng, J., Niu, H., Fang, X., Xu, N., Wu, Z., Guo, S., Wu, Y., Chen, W., Lu, S., Shao, M., Xie, S., Luo, B., and Zhang, Y.: Secondary formation of aerosols under typical high-humidity conditions in wintertime Sichuan Basin, China: a contrast to the North China Plain, J. Geophys. Res-Atmos., 126, D03456, https://doi.org/10.1029/2021jd034560, 2021.

5. Line 396-399. The authors provided a detailed explanation for the synchronous increase of T and RH during the haze period in this study. They believe that the elevated RH caused an increase in T. I am curious why a similar phenomenon did not occur in Beijing, where RH increased but the T decreased.

Response. Thanks for this valuable suggestion. The relationship between meteorological factors and

pollution formation is complex, and there are obvious temporal and spatial differences in their relationship (Zhang et al., 2015; Zhang et al., 2019; Zhang et al., 2014; Yang et al., 2016). Meanwhile, the relationship between T and RH during the pollution process and their mutual influence mechanisms are beyond the scope of this study. Therefore, we have removed the discussion here.

Reference.

Yang, Y., Liao, H., and Lou, S. J.: Increase in winter haze over eastern China in recent decades: Roles of variations in meteorological parameters and anthropogenic emissions, J. Geophys. Res-Atmos., 121, 13,050-013,065, https://doi.org/10.1002/2016jd025136, 2016.

Zhang, R. H., Li, Q., and Zhang, R. N.: Meteorological conditions for the persistent severe fog and haze event over eastern China in January 2013, Sci. China Earth Sci., 57, 26-35, https://doi.org/10.1007/s11430-013-4774-3, 2014.

Zhang, X. Y., Xu, X. D., Ding, Y. H., Liu, Y. J., Zhang, H. D., Wang, Y. Q., and Zhong, J. T.: The impact of meteorological changes from 2013 to 2017 on $PM_{2.5}$ mass reduction in key regions in China, Sci. China Earth Sci., 62, 1885-1902, https://doi.org/10.1007/s11430-019-9343-3, 2019.

Zhang, Z. Y., Zhang, X. L., Gong, D. Y., Quan, W. J., Zhao, X. J., Ma, Z. Q., and Kim, S.-J.: Evolution of surface $O_3$ and $PM_{2.5}$ concentrations and their relationships with meteorological conditions over the last decade in Beijing, Atmos. Environ., 108, 67-75, https://doi.org/10.1016/j.atmosenv.2015.02.071, 2015.

6. Fig. 9/Line 500-503. What are the potential effects of changes in the mixed structure of soot particles. I suggest the authors supplement necessary discussions.

Response. Thanks for this valuable suggestion. In fact, a preliminary discussion has been presented in section 4.2. According to the reviewer's suggestion, we have added a more in-depth discussion (Line 618-625): "*With the aggravation of pollution, soot particles mixed with other particles, and their particle sizes and morphologies undergone significant changes, which will further lead to changes in their hygroscopicity and optical properties, meaning that, ultimately, their climatic and environmental effects may differ significantly from when they exist alone (Adachi et al., 2010; Zhang et al., 2018). For example, Zhang et al. (2023a) found that the formation of organic coatings under the high RH could induce soot redistribution from the particle center to the edge in embedded soot-containing particles compared to partly coated soot-containing particles, and the soot redistribution reduces ~13% optical absorption enhancement of long-range transported soot particles, and the radiative absorption of long-range transported soot particles with a core-shell structure is overestimated by ~20% in the traditional Mie optical model.*"

Reference.

Adachi, K., Chung, S. H., and Buseck, P. R.: Shapes of soot aerosol particles and implications for their effects on climate, J. Geophys. Res-Atmos., 115, D152061, https://doi.org/10.1029/2009jd012868, 2010.

Zhang, Y., Yuan, Q., Huang, D., Kong, S., Zhang, J., Wang, X., Lu, C., Shi, Z., Zhang, X., Sun, Y., Wang, Z., Shao, L., Zhu, J., and Li, W.: Direct observations of fine primary particles from residential coal burning: insights into their morphology, composition, and hygroscopicity, J. Geophys. Res-Atmos., 123, 12964-12979, https://doi.org/10.1029/2018jd028988, 2018.

Zhang, J., Li, W., Wang, Y., Teng, X., Zhang, Y., Xu, L., Yuan, Q., Wu, G., Niu, H., and Shao, L.: Structural collapse and coating composition changes of soot particles during long-range transport, J. Geophys. Res-Atmos., 128, e2023JD038871, https://doi. org/10.1029/2023JD038871, 2023a.

7. Line 524. Please provide a definition of "primary sources", which types of sources does it include? The current expression is unclear.

Response. Thanks for this important comment. According to the comment, we have added the necessary definitions (Line 495-497): "*Based on PMF, there were six factors identified in this study: dust, biomass burning, coal combustion, industrial processes, vehicular emissions and secondary sources (S3). Except for secondary sources, the other five factors can be referred to as primary sources.*"

8. Line 551-553. "During the NP-2 period, with the removal by precipitation and strong easterly winds, the contributions of secondary sources and vehicular emissions decreased by 14.3% and 9.6%, respectively". The significant reduction in secondary sources is easy to understand as they are hygroscopic. Why has the contribution of vehicular emissions also shown a significant decrease? The authors mentioned earlier that the main emission species of vehicular emissions are carbonaceous components, which are weakly hygroscopic or hydrophobic.

Response. This is mainly because Haze-1 was mainly caused by the accumulation of pollutants from vehicular emissions. As show in Fig. 8, compared to NP-1, the proportion of vehicular emissions increased by 19.5% during Haze-1 and reached 38.3%. Therefore, after precipitation and strong winds occurred, these accumulated vehicular emissions pollutants were removed, and their contribution correspondingly decreased. Meanwhile, the strong winds also brought pollutants emitted by other sources (such as coal combustion and dust sources), which further caused a decrease in the relative contribution of vehicular emissions. This explanation has been added to the main text (Line 529-535): "*During the NP-2 period, the precipitation and strong easterly winds not only reduced the contribution of secondary sources rich in hygroscopic species (such as SNA) by 14.3%, but also effectively cleared (decreased by 9.6%) a large amount of accumulated vehicular emissions pollutants during Haze-1. On the contrary, the contributions of coal combustion and dust sources increased by 17.9% and 6.8%, respectively. This is because the continuous easterly winds carried pollutants related to coal combustion in eastern Sichuan Province and Chongqing, and greatly increased the contribution of coal combustion. At the same time, this continuous easterly wind also drove the contribution of dust in NP-2 to reach its highest level across the four periods (15.9%).*"

9. In addition, some comments lack necessary references, such as Line 95-97. The format of references needs to be unified and standardized.

Response. We have checked the entire text and added necessary references (Line 96 and 102; S3 in Supplementary materials). Meanwhile, we have corrected the format of the references based on the "Copernicus_Word_template".